# Pervasive supply of therapeutic lysosomal enzymes in the CNS of normal and Krabbe-affected non-human primates by intracerebral lentiviral gene therapy

Vasco Meneghini[1,†], Annalisa Lattanzi[1,‡], Luigi Tiradani[1], Gabriele Bravo[1], Francesco Morena[2], Francesca Sanvito[3], Andrea Calabria[1], John Bringas[4], Jeanne M Fisher-Perkins[5], Jason P Dufour[5], Kate C Baker[5], Claudio Doglioni[3,6], Eugenio Montini[1], Bruce A Bunnell[5], Krystof Bankiewicz[4,§], Sabata Martino[2], Luigi Naldini[1,6] & Angela Gritti[1,*]

## Abstract

Metachromatic leukodystrophy (MLD) and globoid cell leukodystrophy (GLD or Krabbe disease) are severe neurodegenerative lysosomal storage diseases (LSD) caused by arylsulfatase A (ARSA) and galactosylceramidase (GALC) deficiency, respectively. Our previous studies established lentiviral gene therapy (GT) as a rapid and effective intervention to provide pervasive supply of therapeutic lysosomal enzymes in CNS tissues of MLD and GLD mice. Here, we investigated whether this strategy is similarly effective in juvenile non-human primates (NHP). To provide proof of principle for tolerability and biological efficacy of the strategy, we established a comprehensive study in normal NHP delivering a clinically relevant lentiviral vector encoding for the human ARSA transgene. Then, we injected a lentiviral vector coding for the human GALC transgene in Krabbe-affected rhesus macaques, evaluating for the first time the therapeutic potential of lentiviral GT in this unique LSD model. We showed favorable safety profile and consistent pattern of LV transduction and enzyme biodistribution in the two models, supporting the robustness of the proposed GT platform. We documented moderate inflammation at the injection sites, mild immune response to vector particles in few treated animals, no indication of immune response against transgenic products, and no molecular evidence of insertional genotoxicity. Efficient gene transfer in neurons, astrocytes, and oligodendrocytes close to the injection sites resulted in robust production and extensive spreading of transgenic enzymes in the whole CNS and in CSF, leading to supraphysiological ARSA activity in normal NHP and close to physiological GALC activity in the Krabbe NHP, in which biological efficacy was associated with preliminary indication of therapeutic benefit. These results support the rationale for the clinical translation of intracerebral lentiviral GT to address CNS pathology in MLD, GLD, and other neurodegenerative LSD.

**Keywords** brain; gene therapy; lentiviral vectors; leukodystrophy; non-human primates

**Subject Categories** Genetics, Gene Therapy & Genetic Disease; Neuroscience

## Introduction

Genetic deficiency in the lysosomal enzymes arylsulfatase A (ARSA) and galactosylceramidase (GALC) causes metachromatic leukodystrophy (MLD) and globoid cell leukodystrophy (GLD or Krabbe disease), respectively. Clinical manifestations are neurological with prominent white matter signs in the central and peripheral nervous system (CNS, PNS), reactive astrocytic gliosis, and neuroinflammation (Vellodi, 2005; Kohlschutter, 2013). Treatment options are

---

1   San Raffaele Telethon Institute for Gene Therapy (SR-Tiget), Division of Regenerative Medicine, Stem Cells and Gene Therapy, IRCCS San Raffaele Scientific Institute, Milan, Italy
2   Department of Chemistry, Biology and Biotechnologies, Biochemistry and Molecular Biology Unit, University of Perugia, Perugia, Italy
3   Anatomy and Histopathology Department, San Raffaele Scientific Institute, Milano, Italy
4   University of California San Francisco (UCSF), San Francisco, CA, USA
5   Division of Regenerative Medicine, Tulane National Primate Research Center, Covington, LA, USA
6   Vita-Salute San Raffaele University, Milan, Italy[§]
   *Corresponding author. Tel: +39 02 2643 4623; Fax: +39 02 2643 4668; E-mail: gritti.angela@hsr.it
   [§]Correction added on 2 May 2016, after first online publication: Affiliation 6 was added and the author name was corrected.
   [†]Present address: Institute Imagine, Paris, France
   [‡]Present address: Genethon, Evry Cedex, France

limited, may provide benefit only to some patients, and do not halt all disease manifestations. Allogeneic hematopoietic cell transplantation (HCT) can delay onset or mitigate progression of some CNS manifestations of GLD and MLD when performed in the pre-symptomatic stage (Escolar et al, 2005; Duffner et al, 2009; Krageloh-Mann et al, 2013; Martin et al, 2013). Nevertheless, HCT-treated patients still experience severe neurological defects. As a result, MLD and GLD remain devastating diseases with poor prognosis, especially in the early onset forms.

Transplantation of autologous hematopoietic stem cells (HSC) engineered using lentiviral vectors (LV) to express supraphysiological ARSA levels benefits both CNS and PNS manifestations in MLD patients if performed prior to or very soon after appearance of major symptoms (Biffi et al, 2013). This remarkable advantage is associated with stable reconstitution of ARSA activity to normal levels in the cerebrospinal fluid (Biffi et al, 2013). However, the expected delay in enzymatic reconstitution of CNS tissues by HSC-derived myeloid cells might hamper timely therapeutic advantage of this treatment in symptomatic (even at the early stages) or late-onset patients. Indeed, recent studies showing the synergic effect of intracerebral gene therapy and HCT in the severe and rapidly progressing GLD mouse model clearly show that high enzymatic activity is necessary but not sufficient to counteract the disease progression if not provided in the early asymptomatic stage and with the appropriate timing to all affected tissues (Ricca et al, 2015). In fact, storage, neuroinflammation, CNS, and PNS damage are present well before the onset of symptoms in GLD mice (Meisingset et al, 2013) as well as in GLD and MLD patients (Igisu & Suzuki, 1984; Siddiqi et al, 2006; Biffi et al, 2013; Krageloh-Mann et al, 2013). So, there is a strong rationale for the development of alternative or complementary strategies able to achieve rapid and efficient reconstitution of the functional enzyme in CNS tissues, in order to counteract disease progression or stabilize the pathology. Intracerebral delivery of vectors coding for the functional proteins might fit these requirements.

Intracerebral gene delivery of adeno-associated vectors (AAV) has reached clinical testing for several LSD (Worgall et al, 2008; Leone et al, 2012; Tardieu et al, 2014) and a phase I/II clinical trial evaluating intracerebral AAV-mediated delivery of the ARSA enzyme in late infantile MLD patients has started recently (ClinicalTrials.gov Identifier: NCT01801709). Clinical findings indicate a favorable safety profile of the AAV-mediated approach, while efficacy is overall unclear. Limitations to AVV-mediated GT are related to pre-existing immunity to the viruses (High & Aubourg, 2011; Mingozzi & High, 2013) and size of the transgene that can be delivered (Grieger & Samulski, 2012). LV accommodate larger gene inserts than AAV, display low immunogenicity upon intracerebral (Lattanzi et al, 2014) and systemic delivery (Cantore et al, 2015), and transduce mammalian neuronal and glial cells with high efficiency (Abordo-Adesida et al, 2005; Wong et al, 2006; Lattanzi et al, 2010). The results of the first clinical trial using a LV-based gene therapy vector (ProSavin) to treat chronic Parkinson's disease have been recently published, showing a favorable safety profile and indication of efficacy in all treated patients (Palfi et al, 2014).

Our previous studies on MLD and GLD murine models established the LV-mediated GT platform as a rapid, effective, and safe therapeutic intervention that exploits endogenous neural cells as a long-lasting source of therapeutic enzyme following a single vector injection in white matter tracts (Lattanzi et al, 2010, 2014). Also, they documented lack of adverse effects and absence of insertional genotoxicity in the adult and early postnatal CNS (Lattanzi et al, 2014). Indeed, the continuous improvement in LV design has strongly reduced the risk of insertional mutagenesis related to the use of integrating LV (Montini et al, 2009; Biffi et al, 2011).

While these results highlight a major rationale for LV application in intracerebral gene therapy platforms for LSD, scaling up of the strategy in large animals is required in the perspective of clinical development. Here, we investigated feasibility, safety, and efficacy of intracerebral LV gene therapy to deliver high levels of therapeutic lysosomal enzymes in non-human primates (NHP), in a physiological background and, for the first time ever, in a spontaneous Krabbe NHP model that recapitulates the human pathology.

## Results

### Rationale of the study

We have previously demonstrated comparable patterns of vector biodistribution as well as widespread enzyme expression and activity in CNS tissues upon a single intracerebral injection of LV.ARSA and LV.GALC in white matter (WM) tracts of MLD and GLD mice, respectively, and of WT littermates (Lattanzi et al, 2010, 2014). Here, we sought to assess whether a similar favorable outcome could be achieved upon two injections of therapeutic LV in the brain of non-human primates (NHP), to ultimately provide a rationale for safe and effective clinical development and potential broad application of this platform to address CNS pathology in MLD and GLD. We envisaged targeting the thalamic region in addition to WM tracts (Colle et al, 2010; Rosenberg et al, 2014; Zerah et al, 2015), thus exploiting cortical–spinal tracts and thalamic-cortical connections to enhance vector dispersion and distribution of transgene products from transduced cells. Also, we chose to treat juvenile animals, to mimic the potential treatment of infantile/early juvenile patients. To obtain strong evidence of safety and biological efficacy, we established a comprehensive study in normal NHP, delivering a clinically relevant LV.hARSA manufactured according to the same process used for the current hematopoietic stem cell gene therapy trial of MLD (Biffi et al, 2013). The use of significant sample size ($n = 6$) and the possibility to set up the protocol (i.e. dose, volume and site of injection, enhanced delivery system) in a control group (LV.GFP-injected NHP; $n = 2$) allowed us to evaluate in detail behavior, neuroinflammation, immune response to vector and transgenes, and potential genotoxicity in addition to vector and transgene biodistribution in this experimental setting. Then, in order to collect crucial information regarding the efficacy of this strategy in a severe LSD setting, we took advantage of the unique NHP model of Krabbe disease, which has never been used so far to test therapeutic approaches. The limited availability of Krabbe NHP ($n = 1$) and non-affected controls ($n = 1$), and some technical issues related to their use reduced the possibility to evaluate different experimental conditions and imposed a slightly modified (and perhaps suboptimal) protocol in terms of LV.hGALC dose and delivery strategy as compared to that described for LV.hARSA injec-

tions. Due to these differences in the procedures, results related to LV.ARSA and LV.GALC treatments are presented separately, but with a focus on common patterns and final outcome, with the aim of supporting reliability and robustness of the proposed LV-mediated gene therapy platform and providing for the first time ever preliminary evidence of its therapeutic efficacy in the exclusive NHP model of LSD.

## Feasibility, biological efficacy, and safety profile of LV intracerebral injections delivering the hARSA enzyme in juvenile normal NHP

### Safe and consistent delivery of LV.hARSA in normal NHP

We used normal juvenile NHP (*Macaca fascicularis*; age 2–3 months; $n = 9$; Appendix Table S1), which are less restricted to HIV as compared to other macaques and thus provide a suitable model to asses LV gene transfer before human testing. Importantly, we used a clinically relevant LV.hARSA batch manufactured according to the same process used for the *ex vivo* HSC gene therapy trial of MLD (Biffi *et al*, 2013) (Appendix Table S2) and validated for efficacy and safety in MLD mice (As2$^{-/-}$) (Appendix Fig S1). We used convection enhanced delivery CED (Lonser *et al*, 2015) with the aim of obtaining reproducible delivery of vector suspension while enhancing LV particle dispersion. Since no published data were available regarding CED-mediated LV intracerebral injection in juvenile NHP, we optimized surgical procedures by using two NHP (P1 and P2; pilot group), which were injected unilaterally in the EC and thalamus (Fig 1A) with different doses/volumes of LV.GFP (Table 1) and different CED parameters (Appendix Table S3). Based on the results collected from these pilot animals, we decided to inject $5 \times 10^7$ TU/injection site (80 μl) of LV.hARSA (Table 1), a combination that allowed limiting the total time of surgery (Appendix Table S3) without affecting the volume of injected suspension, as assessed by 3D rendering of injected brains generated from post-surgery MRI scan (Fig 1B and Appendix Table S4). According to data showing increased GFP expression in the caudal brain regions and spinal cord upon targeting posterior WM tracts in mice (A. Gritti, unpublished data), we randomly assigned the 7 NHP to two study groups (study group 1, $n = 3$; study group 2, $n = 4$) based on the site of the second injection (thalamus and posterior EC, respectively) (Table 1 and Fig 1A and B). One NHP assigned to study group 2 died before surgery due to complication of anesthesia (Appendix Table S1), and thus, we treated 3 NHP/group. The vital signs (blood pressure, heart rate, O$_2$ saturation, body temperature) monitored during surgery were within the normal range. The volume of injected vector suspension estimated from 3D brain rendering confirmed the consistency of the injection procedure (Fig 1B and Appendix Table S4). Behavior, motricity (Appendix Fig S2A), as well as body weight (Appendix Fig S2B) and clinical chemistry parameters (Appendix Fig S2C) of treated NHP remained within the normal range during the 3-month follow-up. These results suggested that CED-mediated LV intracerebral injection in juvenile NHP was well tolerated with no serious adverse events. At the end of experiment, animals were perfused with saline. Brain, cervical spinal cord (Fig 1C and D) sciatic nerves, liver, spleen, and gonads were collected and divided into blocks that were either frozen or fixed in PFA. All brain slices and organs presented no macroscopic lesions.

### Efficient LV-mediated transduction of neurons, astrocytes, and oligodendrocytes

Indirect immunofluoroscence (IF) assay followed by confocal analysis showed relevant numbers of GFP$^+$ cells within and around the injection sites of LV.GFP-treated NHP (Fig 2A). We estimated the percentage of transduced cells by GFP labelling and nuclear counterstaining in serial tissue slices encompassing the injection area. A total of 445 nuclei and 173 GFP$^+$ cells (38%) were counted in four slices derived from two tissue blocks. This percentage was comparable to that scored after LV injection in the murine CNS (Lattanzi *et al*, 2010). Quantitative analysis of the transduced cell types showed that $50.5 \pm 5.7\%$ of the GFP$^+$ cells were neurons (NeuN), $22.3 \pm 5.7\%$ were astrocytes (GFAP), and $24.4 \pm 10.6\%$ were oligodendrocytes (CNPase) (mean $\pm$ SEM; $n = 2$, 2–4 slices/brain) (Fig 2A). A similar pattern of cell transduction was observed in LV.hARSA-injected NHP, in which numerous neurons and oligodendrocytes overexpressing the ARSA protein were detected around the injection sites (Fig 2B and C and Appendix Fig S3). Physiological ARSA expression was barely detectable in IF analysis (Fig 2D, Appendix Fig S3). These data indicate that LVs proficiently transduced neurons, astrocytes and oligodendrocytes upon intraparenchymal injection in the juvenile NHP brain, resulting in robust transgene expression.

### Spread of LV particles along white matter tracts and extensive hARSA distribution in CNS tissues

We measured integrated LV genome, transgene expression (GFP and ARSA; mRNA and protein) in tissue blocks cut from brain, spinal cord, PNS, and organ specimens of LV-injected NHP.

The VCN ranged between 0.1 and 2.25 for diploid genome with no differences related to treatment (Table 2), with maximum values in close correspondence to the injection sites (Fig 3, Appendix Figs S4 and S5). By evaluating the number of tissue blocks containing detectable integrated LV genome (VCN > 0.001) in all the slices, we estimated a global vector diffusion of 24–48 mm along the anterior–posterior axis and 20–30 mm along the medial–lateral and dorsal–ventral axes, respectively (Table 2). The significant increase in the volume of transduced hemisphere observed in NHP of the study group 2 (Table 2) suggested more efficient dispersion of LV particles along fibers in the EC system as compared to the thalamus. Vector diffusion in the contralateral hemisphere was minimal, and we did not detect LV genome in spinal cord tissues of all injected NHP. Importantly, VCN values below the threshold of detection were measured in sciatic nerves and in liver, spleen, and gonads of all the injected animals. These data suggested robust LV transduction around sites of injection, moderate diffusion of LV particles, preferentially along WM tracts, and the absence of off-target LV integration.

Western blot (WB) and immunohistochemistry (IHC) analysis on tissues of pilot NHP showed robust and widespread GFP expression along the rostrocaudal axis, in both cell bodies and processes, reaching the cervical spinal cord (Fig EV1). Tissue blocks with detectable GFP expression corresponded to $\approx 50\%$ (P1) and $\approx 70\%$ (P2) of the injected hemisphere volume and to $\approx 24\%$ (P1) and $\approx 45\%$ (P2) of the contralateral hemisphere volume, indicating that two unilateral injections are sufficient to ensure protein transport far from the injected areas in the juvenile NHP brain.

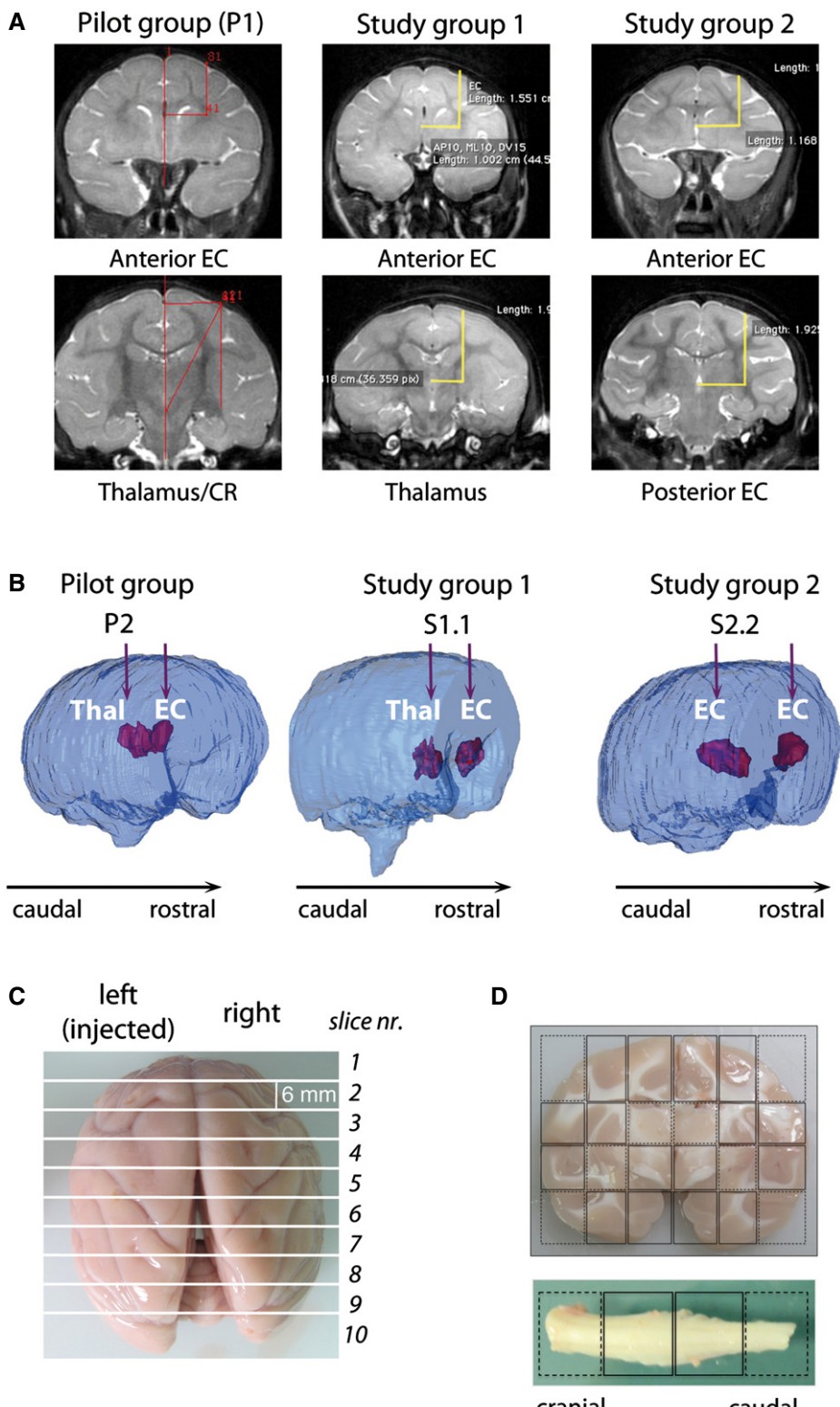

**Figure 1.  LV.GFP- and LV.hARSA-injected NHP: treatment groups and tissue collection.**

A   Coordinates for injection in normal NHP of the pilot group, study group 1 and study group 2, were calculated on the bases of pre-surgery MRI scans.

B   3D rendering of injected brains generated based on post-surgery MRI scans showing the injection sites and the brain volume containing the vector suspension in the different treatments groups. EC, external capsule; Thal, thalamus; CR, corona radiata; ant, anterior; post, posterior.

C   The brain was cut in 6-mm-thick slices using an adjustable brain matrix, obtaining 9–10 slices/brain.

D   Each brain slice was divided along the midline and each part subdivided into blocks. The cervical spinal cord was cut in four blocks.

**Table 1.  Experimental groups and study design.**

| ID | Experimental group | Vector | Target sites (injected volume in μl) | | Total TU/site | | Total TU/brain | Time of analysis (days post-injection) |
|----|----|----|----|----|----|----|----|----|
| P1 | Pilot | LV.GFP | Anterior EC (40) | Thal/CR (80) | $2 \times 10^7$ | $4 \times 10^7$ | $6 \times 10^7$ | 35–36 |
| P2 | | | Anterior EC (150) | Thal/CR (150) | $4 \times 10^7$ | $4 \times 10^7$ | $8 \times 10^7$ | |
| S1.1 | Study group 1 | LV.hARSA | Anterior EC (80) | Thal (80) | $5 \times 10^7$ | | $1 \times 10^8$ | 92–94 |
| S1.2 | | | | | | | | |
| S1.3 | | | | | | | | |
| S2.1 | Study group 2 | | Anterior EC (80) | Posterior EC (80) | $5 \times 10^7$ | | $1 \times 10^8$ | 98–100 |
| S2.2 | | | | | | | | |
| S2.3 | | | | | | | | |

The indicated doses and volumes of LV.GFP suspension were injected via CED in the anterior external capsule (EC) and in the thalamus (Thal)/corona radiata (CR) of the two NHP in the pilot group. The remaining NHP were then randomly assigned to the study groups 1 and 2. A total dose of $1 \times 10^8$ TU of LV.hARSA was injected in two injection sites in NHP of study group 1 (anterior EC and Thal) and study group 2 (anterior and posterior EC). The time of analysis for each group is shown in the last column.

In line with VCN distribution, the expression of transgenic hARSA mRNA was high close to the injection sites and decreased in distant regions of the injected hemisphere in LV.hARSA-injected NHP (Appendix Fig S5). Accordingly, total ARSA mRNA expression reached > 100-fold the physiological levels close to the injection sites (assessed by comparing the region-matched contralateral tissue blocks) (Fig 3). IHC analysis confirmed an overall 50% increase of ARSA expression in the injected hemisphere of LV.hARSA-injected NHP as compared to physiological expression (detected in pair-matched blocks of LV.GFP-injected NHP) (Fig 4). The overexpression was less evident in the contralateral hemisphere, due to the difficulty of distinguishing the likely moderate increase of immunoreactive signal consequent to hARSA transport and uptake over the physiological ARSA expression.

*Supraphysiological levels of ARSA activity in the whole CNS of LV.hARSA-injected NHP*

Analysis performed in a relevant and consistent number of blocks in all brain slices of 2 representative NHP (S1.2 and S2.2) and in selected blocks/brain slices of other NHP showed up to twofold increase of ARSA activity as compared to physiological levels (measured in matched blocks of UT and LV.GFP-injected NHP) (Fig 5A), not only close to the injection sites, as anticipated by IHC data, but also in distant regions of the contralateral hemisphere (Fig 5B and C), with a significant enhancement in NHP receiving the second injection in the posterior EC as compared to those injected in the thalamus (Fig 5D). Supraphysiological (> 20% over the basal levels) ARSA activity was present in ≈ 80% of blocks analyzed in the injected hemisphere of NHP in both study groups and in ≈ 20 and ≈ 50% of blocks analyzed in the contralateral hemisphere of NHP in study group 1 and study group 2, respectively. Importantly, a higher percentage of blocks with ARSA activity > 30 and > 50% over the physiological levels was present in the contralateral hemispheres of NHP in study group 2 when compared to study group 1 (Fig EV2A).

The transgenic hARSA protein reconstituted in the brain of injected NHP showed biochemical features indistinguishable from those of the native enzyme (Fig EV2B; chromatographic analysis) and displayed efficient catalytic activity toward the natural substrate (sulfatide) (Morena *et al*, 2014) (Fig EV2C). A trend for increased

ARSA activity was also observed in the spinal cord of LV.hARSA-injected NHP (Fig 5E). No detectable increase of ARSA above normal levels was observed in the sciatic nerve of both groups (Fig 5F). The twofold increase of ARSA activity measured in the cerebrospinal fluid (CSF) of LV.hARSA-injected NHP (Fig 5G) indicated enzyme release and transport in the liquor and pointed to long-term persistence of functional enzyme expression from transduced cells due to the treatment.

*Favorable safety profile of intracerebral LV.hARSA gene therapy*

Histopathological evaluation was made on all available brain slices and on organ specimens, and a severity score was attributed according to size and number of observed lesions, which included perivascular mononuclear infiltration, gliosis, and mononuclear infiltration in the adjacent neuropil (Appendix Table S5). Histopathological lesions were observed exclusively in close proximity of the injection sites of five out of eight animals, including one animal with severe, two with moderate, and two with mild inflammatory lesions (Appendix Table S5). Infiltrates were mainly composed by CD3[+] and CD20[+] cells (T and B lymphocytes), and CD11c[+] macrophages/dendritic cells (Fig EV3A and B). The presence of infiltrating macrophages was confirmed by mRNA expression of chemokine (C-C motif) ligand 2 (CCL2), which correlated with histology and IHC data (Fig EV3C). Few cells with neuronal appearance displaying homogenous eosinophilic cytoplasm (interpreted as intracytoplasmic protein accumulation) and lateral displacement of nucleus were found in gray matter regions close to injection sites (Appendix Table S5 and Fig EV3D). All remaining brain specimens, including matched blocks of the contralateral hemisphere (Fig EV3B), and all examined organs (sciatic nerves, liver, spleen, and gonads) presented no detectable histologic lesions.

We found a low titer of antibodies against LV particles in 2 out of 8 animals (Fig EV3E and F). Importantly, we never found p24 antigen in sera collected at the end of experiment, indicating the absence of replication-competent LV particles in the circulation. Low titer of antibodies against GFP was present in one of the pilot NHP (Fig EV3G), while no antibodies against hARSA were found in LV.hARSA-injected NHP.

In order to assess a potential genotoxic effect of LV-mediated gene transfer at the molecular level, we characterized the genomic

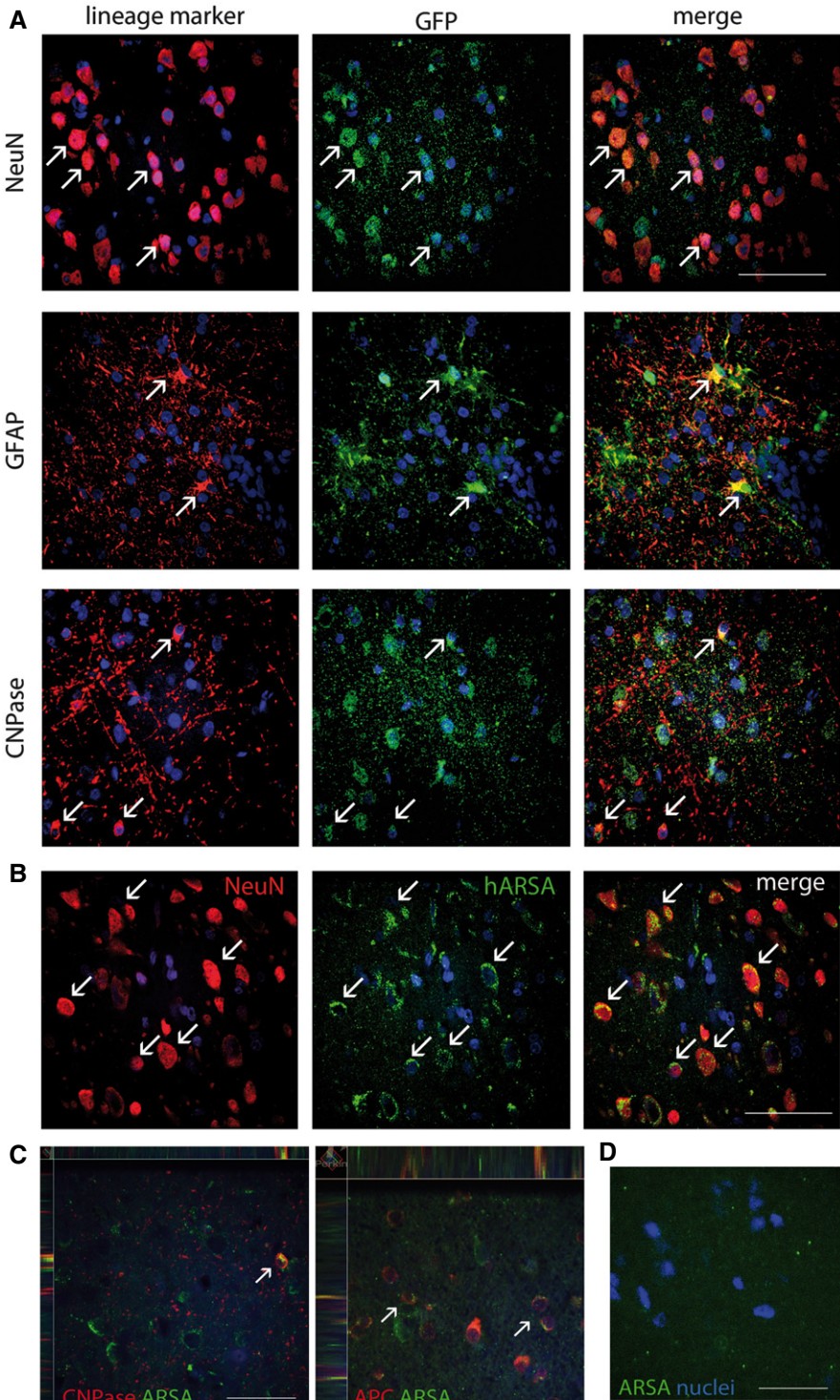

**Figure 2.   LV transduction and transgene overexpression in neurons and glial cells at the injection sites.**

A   Representative confocal images showing GFP+ cells (green) expressing markers of neurons (NeuN), astrocytes (GFAP), and oligodendrocytes (CNPase) close to the posterior injection site of P2 NHP. Arrows indicate co-localization of IF signals.
B   Representative confocal images showing neurons (NeuN) overexpressing ARSA (green) close to the posterior injection site of S2.2 NHP. Arrows indicate co-localization of IF signals. Note the granular perinuclear ARSA staining, likely representing protein localized in the Golgi/lysosomal vesicles.
C   Confocal z-stack images showing the presence of ARSA-positive cells (green) co-stained with oligodendrocyte markers CNPase and APC (arrows).
D   Undetectable ARSA immunoreactivity in tissues of untreated animals.

Data information: Scale bars: 80 μm (A, C), 60 μm (B, D). In all pictures, nuclei are counterstained with ToPro (blue).

**Table 2. Diffusion of LV particles in the NHP brains and estimate of transduced brain volume.**

| NHP | Experimental group | VCN at inj. sites 1st/2nd | Diffusion of LV particles (mm) | | | % of hemisphere volume containing vector | | % of total brain volume containing vector |
|-----|-----|-----|-----|-----|-----|-----|-----|-----|
| | | | AP | ML | DV | Injected | Contralateral | |
| P1 | Pilot group | 1.17/1.62 | 30 | 30 | 24 | 13 | 5 | 9 |
| P2 | | 0.58/0.48 | 48 | 20 | 20 | 16 | 1 | 8.5 |
| | Mean ± SEM | 0.96 ± 0.27 | | | | 14.5 ± 1.5 | 3 ± 2 | 8.7 ± 0.25 |
| S1.1 | Study group 1 | 0.67/1.15 | 24 | 20 | 20 | 12 | 2 | 7 |
| S1.2 | | 2.25/0.18 | 36 | 20 | 20 | 8 | 2 | 5 |
| S1.3 | | 0.25/1.46 | 24 | 30 | 30 | 13 | 0 | 6.5 |
| | Mean ± SEM | 0.99 ± 0.32 | | | | 11.0 ± 1.5 | 1.3 ± 0.7 | 6.2 ± 0.6 |
| S2.1 | Study group 2 | 0.37/0.49 | 42 | 20 | 30 | 20 | 1 | 10.5 |
| S2.2 | | 0.75/1.52 | 30 | 20 | 30 | 21 | 1 | 11 |
| S2.3 | | 0.13/0.13 | 24 | 30 | 20 | 15 | 3 | 9 |
| | Mean ± SEM | 0.56 ± 0.21 | | | | 18.7 ± 1.8* | 1.7 ± 0.7 | 10.2 ± 0.6* |

The table shows the vector copy number (VCN) at injection sites, the diffusion of LV particles along the anterior–posterior (AP), medial–lateral (ML), and dorsal–ventral (DV) axes, the percentage of hemisphere volume containing integrated LV in the injected and contralateral hemispheres and in the total brain of NHP in the pilot group and in study groups 1 and 2. Data are presented as mean values ± SEM. Data are analyzed by Kruskal–Wallis test followed by Dunn's multiple comparison tests. *$P < 0.05$ versus study group 1.

integration profile of the LV constructs used in this study in brain tissues of 4 treated NHP (P1, P2, S1.3, and S2.2). We amplified the vector–genome junctions by linear amplification-mediated (LAM)-PCR (Paruzynski *et al*, 2010). The LAM-PCR products sequenced by 454 pyrosequencing were mapped on the *Macaca Fascicularis* genome using a dedicated bioinformatics pipeline. Overall, we retrieved > 4,000 unique integration sites. The proportion of sequencing reads representing each integration site within each data sets (surrogate readout for the relative abundance of vector-marked cell clones in brain tissues of LV-injected NHP) showed a pattern of polyclonal marking in all the four tissue samples analyzed without evidence for expanded clones within the transduced brain (Fig EV3H). In agreement with previous reports (Bartholomae *et al*, 2011; Lattanzi *et al*, 2014), we found that about 60% of LV integrations mapped within genes (Fig EV3I). Of importance, Gene Ontology analysis showed preferential targeting of genes related to neural/neuronal function, without evidence for *in vivo* selection of insertions occurring at cancer genes (Appendix Table S6). Overall, these results indicated a favorable safety profile of intracerebral LV gene therapy in juvenile NHP.

**Therapeutic potential of LV-mediated intracerebral gene therapy in the Krabbe NHP model**

*Intracerebral LV.hGALC delivery is well tolerated in the Krabbe NHP*
While there are no large animal models of MLD, a mutation causing globoid cell leukodystrophy (GLD) in the rhesus monkey has been previously described (Luzi *et al*, 1997). The rhesus macaque model of Krabbe disease has a dinucleotide deletion that abolishes galacto-sylceramidase (GALC) activity and shows clinical signs (muscle tremors of head and limbs, ataxia, hypertonia, and incoordination) and immunopathologic alterations (central and peripheral demyelination, the presence of multinucleated globoid cells, and psychosine accumulation) resembling those found in humans (Baskin *et al*, 1998; Weimer *et al*, 2005; Borda *et al*, 2008). While these animals

represent a unique LSD model to provide proof of principle of safety and efficacy of gene therapy platforms, their limited availability hampers the use of a large sample size to test different therapeutic approaches. We had the opportunity to investigate for the first time LV-mediated intracerebral gene delivery in one Krabbe-affected juvenile NHP and in one normal animal from the same colony. We administered a LV encoding for the hGALC cDNA tagged with Myc peptide (Appendix Table S2), which we tested for efficacy in GALC-deficient cells *in vitro* and upon intracerebral injection in GALC-deficient Twitcher mice (Appendix Fig S6). With the aim of matching the age of LV.hARSA-injected NHP and because of the rapid disease progression in Krabbe-affected animals (average life span 100 days), a young affected animal (JT02; 53 days) was chosen for the study. The normal NHP (JV02) was 89 days old at the time of treatment (Appendix Table S7). We applied a protocol similar to that described for LV.hARSA injection in normal NHP. Since a controlled CED-mediated system was not available, we used a standard Hamilton-driven injection, splitting the total injected volume in multiple deposits. We performed two unilateral injections targeting the internal capsule (three deposits; $5 \times 10^6$ TU/20 μl/deposit) and the thalamus ($2 \times 10^7$ TU/40 μl). The total injected LV.hGALC dose in JT02 and JV02 was $0.35 \times 10^8$ TU/brain as compared to $1 \times 10^8$ TU/brain in LV.hARSA-injected NHP.

Prior to treatment (at 7, 14, and 28 postnatal days), both animals underwent neurobehavioral assessment using a Neurobehavioral Assessment Scale (NBAS) (Brazelton, 1973) standardized for use with juvenile rhesus macaques (Champoux *et al*, 1994, 1997, 2002) that examines central aspects of juvenile's neurobehavioral performance including orientation, state control, motor maturity, and reflexes. With the exception of the 7-day testing period, scores for study animals fell within historical ranges for genetic status and age. In addition, JT02 scored lower than the control animals for most measures and most time points, including motor maturity, activity, neuromotor items, and muscle tone items (Fig EV4A and Appendix Table S8).

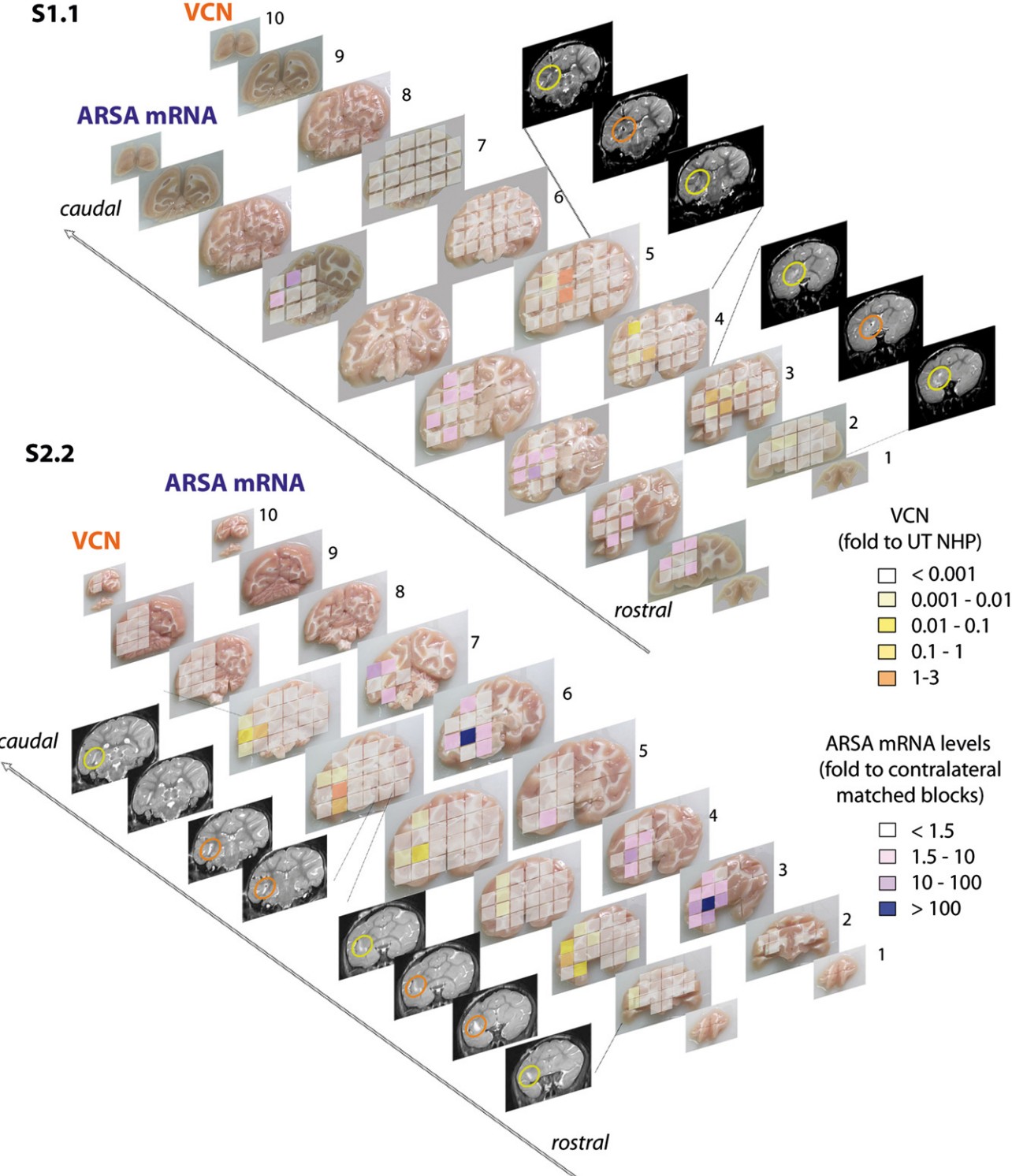

**Figure 3.    Integrated LV genome and transgene overexpression in LV.hARSA-injected NHP.**

Vector copy number (VCN) cartography shows integrated LV genome (assessed by qPCR) along the rostrocaudal axis (slices 1–10) in LV.hARSA-injected S1.1 and S2.2 NHP in a side-by-side comparison with ARSA mRNA expression (assessed by qPCR analyses using probe and primers annealing to sequences conserved in both the human and *Macaca fascicularis* and expressed as fold increase to region-matched blocks of the contralateral hemisphere). Grading of colors for VCN ranged from white (VCN < 0.001; corresponding to CT > 37) to dark orange (VCN = 1–3). The highest VCN is found in close correspondence to the injection sites, as confirmed by comparison with post-surgery MR images (yellow and orange circles indicate viral suspension close to the injection sites). Grading of colors for ARSA expression ranges from white (< 1.5-fold the physiological level) to dark purple (100-fold the physiological level).

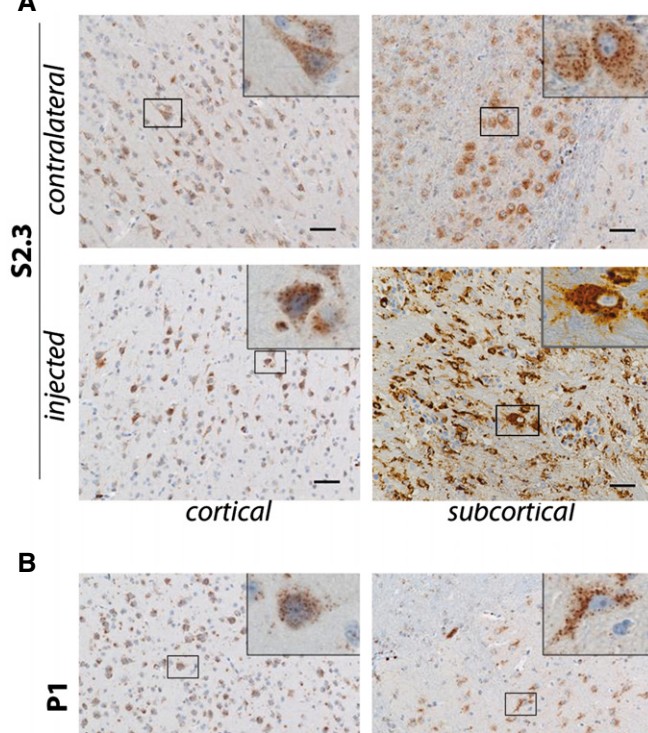

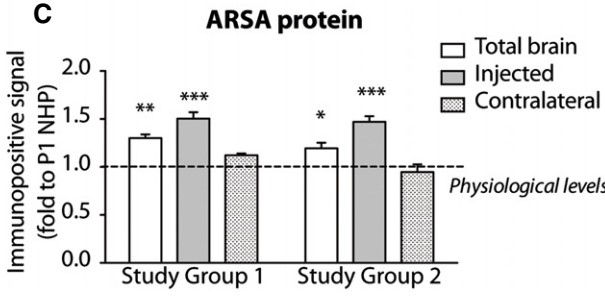

## C

**ARSA protein**

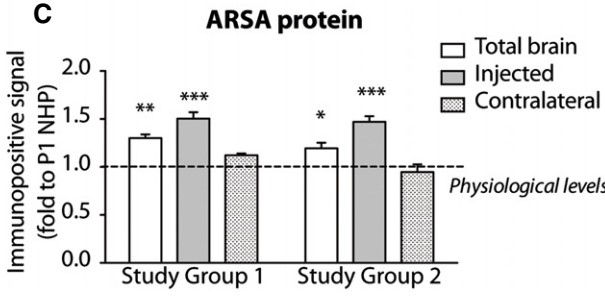

**Figure 4. Supraphysiological ARSA protein expression in LV.hARSA-injected NHP.**

A  ARSA overexpression detected by IHC in cortical and subcortical regions of the injected and contralateral hemispheres of S2.3 NHP. Scale bars, 20 μm.

B  Physiological ARSA expression in cortical and subcortical regions of the injected hemisphere in LV.GFP-injected P1 NHP. Scale bars, 20 μm. Insets show ARSA-expressing cells at higher magnification. Note the perinuclear accumulation of ARSA protein.

C  Quantification of immunopositive signal in brain slices 3–6 (*n* = 8 blocks/ slice). Data are the mean ± SEM. Data are expressed as fold increase to mean immunopositive signal measured in matched brain regions of P1 NHP (physiological levels). Two-way ANOVA and Bonferroni's post-tests, *$P$ < 0.05, **$P$ < 0.01, ***$P$ < 0.001 versus physiological levels.

The surgical procedure was well tolerated with no serious adverse events. All physiological parameters were normal during surgery and in the post-surgery follow-up. Body weight of treated animals was comparable to historical ranges from animals of similar genetic

status and age (Fig EV4B). These results closely resembled those obtained in age-matched LV-injected normal NHP.

Three months after treatment, animals were perfused with saline. Tissues (CNS and organ specimens) were collected to assess pathology, vector, and transgene biodistribution, as well as specific GALC activity (Appendix Fig S7).

We documented moderate astrogliosis and microgliosis in correspondence to the injection sites in JV02 (Fig EV5A and B). This local reaction was not easily assessable by immunofluorescence (IF) in JT02, since Krabbe-affected monkeys show diffused astrogliosis and microgliosis in several brain regions (Borda *et al*, 2008). Increased GFAP and Iba-1 expression assessed by WB analysis (Fig EV5B) and upregulation of chemokine CCL2 mRNA expression levels in regions close to the injection sites of JT02 as compared to untreated Krabbe NHP (Fig EV5C) suggested the occurrence of local inflammation and macrophage recruitment as a consequence to the surgical procedure. Tumor necrosis factor-α (TNF-α) and interleukin-1β (IL-1β) mRNA expression levels were increased in JT02 tissues when compared to physiological levels but were not significantly altered as a consequence of LV injection, as assessed by comparison with tissues from untreated Krabbe-affected animals (Fig EV5D and E).

### Local LV.hGALC-mediated transduction establishes widespread distribution of a functional GALC enzyme in Krabbe-affected CNS tissues and CSF

We found integrated LV genome in proximity to the injection sites (slices 9–13; VCN range: 0.00013–0.061 for diploid genome) (Fig 6A). The lower VCN values as compared to those measured in LV.GFP- and LV.hARSA-injected NHP (Table 2) likely reflect the lower injected LV dose that, coupled to the limited particle dispersion resulting from non-CED-mediated delivery, ultimately determined the volume of injected hemisphere containing integrated LV, which was ≈ 3% as compared to 11–18% in LV.hARSA-injected NHP (Table 2). GALC mRNA expression levels ranged between twofold and eightfold the physiological levels (assessed by comparing region-matched contralateral tissue blocks) (Fig 6B), in close correlation with VCN values. Integrated LV genome and hGALC mRNA were undetectable in spinal cord, sciatic nerves, liver, spleen, and gonads of both JT02 and JV02, confirming the absence of off-target LV integration upon injection in normal and Krabbe-affected brains.

Importantly, by IF staining followed by confocal analysis, we showed relevant numbers of GALC-expressing neurons and glial cells within and around the injection sites of LV.hGALC-injected NHP (Fig 6C), as well as in matched blocks of the contralateral hemisphere up to ≈ 2 cm rostral and caudal from the injection site (Fig 6D and E).

Taken together, results obtained in LV.GFP-, LV.hARSA-, and LV.hGALC-injected NHP showed a reproducible and transgene-independent pattern of tissue transduction and vector biodistribution upon intracerebral LV delivery. Specifically, a small pool of LV.hGALC-transduced cells efficiently produced and released the hGALC protein, which was available for cross-correction of surrounding GALC-deficient cells in the context of a disease-affected brain.

In order to evaluate the biological efficacy of the treatment, we measured the specific GALC activity (Martino *et al*, 2009) in a

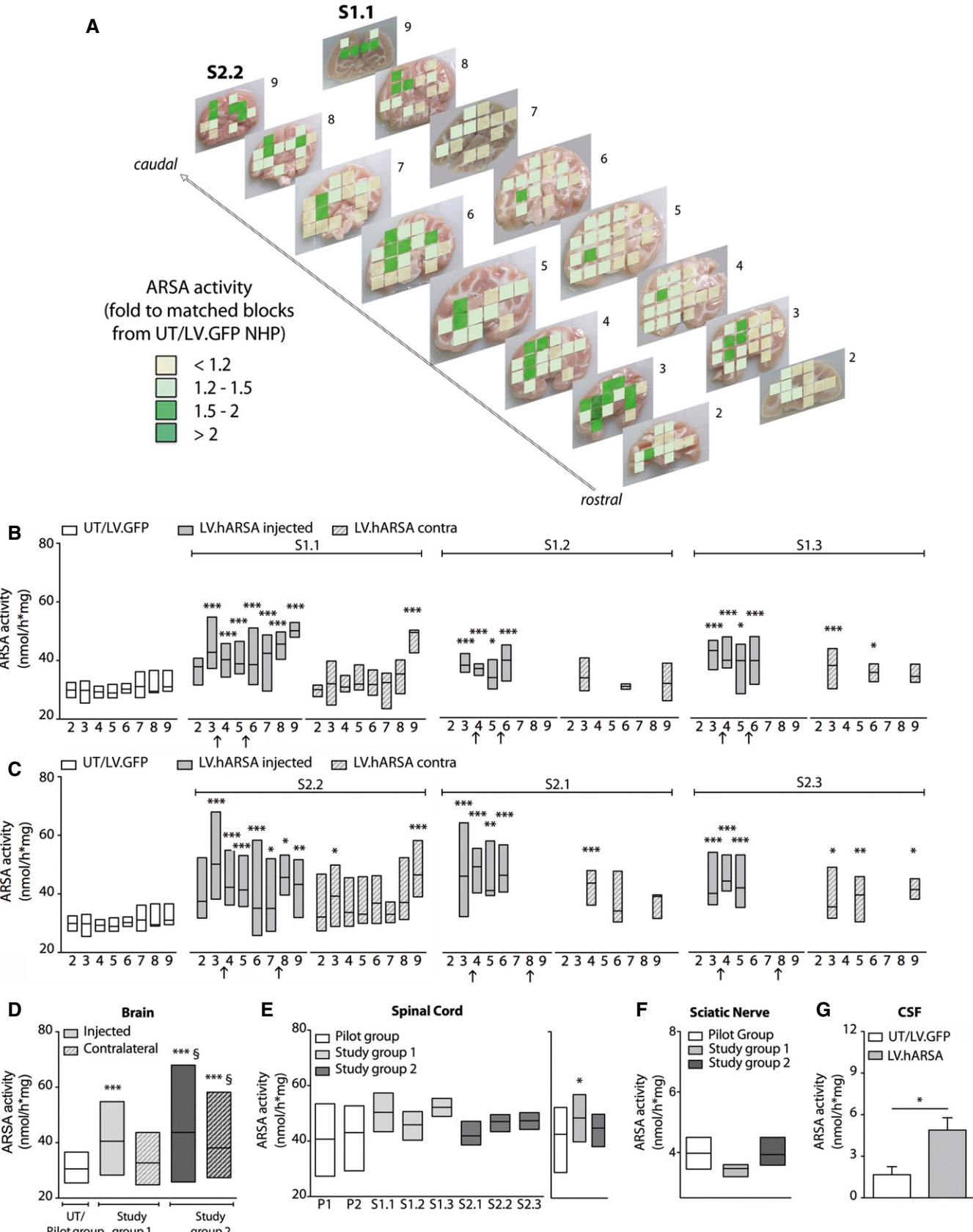

Figure 5.

◄

**Figure 5.  Supraphysiological ARSA activity in LV.hARSA-injected NHP.**

A    Schematic representation of ARSA activity in the brain of S1.1 and S2.2 NHP along the rostrocaudal axis (slices 2–9). Data are expressed as fold increase to ARSA activity detected in matched blocks of UT and LV.GFP-injected NHP (physiological levels). Grading of colors ranges from light yellow (< 1.2-fold) to dark green (≥ twofold).

B, C  Enzymatic activity measured in single brain slices of NHP of study group 1 (B) and study group 2 (C) (filled gray bars, injected hemisphere; striped gray bars, contralateral hemisphere) and in UT and LV.GFP-injected NHP (white bars; physiological levels). Arrows on *x*-axis indicate the injection sites; *n* = 2–12 tissue blocks/slice. Number of tissue blocks analyzed per animal: 51 (UT/LV.GFP), 98–122 (S1.1 and S2.2), and 12–22 (S1.2, S1.3, S2.2, S2.3). Two-way ANOVA followed by Bonferroni's multiple comparison tests, *$P < 0.05$; **$P < 0.01$; ***$P < 0.001$ versus matched slices of UT/LV.GFP.

D    Summary of ARSA activity in the injected and contralateral hemispheres of NHP in study group 1 and study group 2. Number of tissue blocks analyzed were as follows: 51 (UT/LV.GFP), 105 (study group 1, injected hemisphere) and 52 (study group 1, contralateral hemisphere); 103 (study group 2, injected hemisphere) and 72 (study group 2, contralateral hemisphere). One-way ANOVA and Tukey's multiple comparison tests, ***$P < 0.001$ versus UT/LV.GFP, $^{\S}P < 0.05$ versus study group 1.

E    ARSA activity in the spinal cord of individual LV.GFP- (*n* = 4 blocks/animal, 2–3 replicates) and LV.hARSA-injected NHP (*n* = 2–4 blocks/animal; *n* = 10 and *n* = 11 total blocks for study group 1 and study group 2, respectively). Graph on the right represents the summary of ARSA activity in each treatment group. One-way ANOVA and Dunnet's multiple comparison test. *$P < 0.05$ versus pilot group.

F    ARSA activity in the sciatic nerve of LV.GFP- (*n* = 2) and LV.hARSA-injected NHP (*n* = 3). In (B–F), data are expressed as floating bars (line at mean).

G    ARSA activity in the cerebrospinal fluid (CSF) of UT/LV.GFP (*n* = 3) and LV.hARSA-injected NHP (*n* = 5). Data are expressed as mean ± SEM. Student's *t*-test, *$P = 0.035$.

relevant number of blocks in brain slices and in spinal cord tissues of JT02 and JV02, as well as in tissues of age-matched untreated Krabbe-affected and normal NHP. Tissues from untreated Krabbe-affected NHP (*n* = 4; Appendix Table S7) showed undetectable GALC activity. In contrast, ≈ 70% of physiological GALC activity was measured in the injected and contralateral brain hemispheres (Fig 7A and B), and ≈ 40% of physiological GALC activity was measured in spinal cord tissues (Fig 7C) of JT02. GALC overexpression (+30–40% over physiological level) was observed in brain tissues of JV02 (Fig 7A and B). The chromatographic profile of GALC (Lattanzi *et al*, 2010) obtained from selected brain slices and spinal cord of JT02 and untreated non-affected NHP demonstrated the comparable native conformation of transgenic and WT GALC (Fig 7D). Importantly, GALC activity in the CSF of JT02 reached ≈ 10% of that measured in JV02 (Fig 7E), which likely reflected supraphysiological levels of GALC activity, as also detected in JV02 CNS tissues (Fig 7A and B). These data suggested that LV-mediated gene therapy supplies therapeutically relevant levels of functional hGALC enzyme in CNS tissues and CSF of Krabbe NHP.

*Preliminary indication of safety and therapeutic efficacy of intracerebral LV.hGALC gene therapy*

Taking advantage of the unique Krabbe's behavioral manifestations, we sought to obtain descriptive information on response to treatment as a preliminary evaluation of safety and therapeutic efficacy of the LV gene therapy strategy. To this end, JT02 and JV02 were evaluated monthly (beginning at 2 months of age) using a modified Bayley's scale for infant development that included problem-solving, motor ability, and temperament tests (Champoux *et al*, 1994). Results were compared with historical data relative to age- and genotype-matched juveniles. JT02 had one assessment prior to

surgery (2 months) while JV02 had 2 pre-surgery assessments (2 and 3 months). The control juvenile JV02 was uncooperative, irritable, and hostile during testing and often would make no attempt to complete items presented, resulting in unusually low scores on the cognitive subtest and high scores on the behavior/social orientation subtest (Appendix Table S9), while motor scores fell within the historical range (Appendix Table S9 and Fig 7F). Interestingly, JT02 showed unprecedentedly high motor performance scores in the post-surgery assessment when compared to untreated Krabbe-affected juveniles. This score measures muscle tone and strength of muscle contractions (i.e. resistance to the experimenter gently flexing the limbs) as well as righting reflexes (i.e. reorienting the body when shifted out of normal position by the experimenter). JT02's scores showed the age-dependent increase seen in normal juveniles, and never seen in Krabbe-affected juveniles, which show instead an age-dependent decrease. Indeed, while JT02's motor scores fell below normal animals and within range for affected juveniles at 2–4 months, JT02's score was higher than the score of any prior Krabbe-affected juvenile at 5 months (3 months after gene therapy), at a level that fell within normal range and with a mean score almost identical to that of normal age-matched juveniles (Fig 7F).

Overall, these data provide the first preliminary evidence of safety and therapeutic benefit of LV.hGALC gene therapy in Krabbe NHP.

## Discussion

Here, we show that lentiviral vector-mediated intracerebral gene therapy in juvenile NHP displays a favorable safety profile and

**Figure 6.  Efficient transduction of Krabbe CNS tissue by LV.hGALC.**

A    Vector copy number (VCN) indicating the distribution of integrated LV genome along the rostrocaudal axis (slices 7–19) in LV.hGALC-injected NHP (JT02 and JV02) assessed by qPCR. Each dot represents the VCN measured in one block within the slice. Lower threshold (dotted line): VCN < 0.001, corresponding to CT > 37.

B    GALC mRNA expression along the rostrocaudal axis of JT02 (slices 7–13) is expressed as fold increase to region-matched blocks of the contralateral hemisphere (dotted line, y = 1). Data are expressed as floating bars (min to max, line at mean; *n* = 4–8 blocks/slice). Arrows on *x*-axis in (A) and (B) indicate the injection sites.

C    Confocal images showing GALC⁺ cells expressing markers of neurons (NeuN) and astrocytes (S100β) close to the injection site (slice 10) in JT02. Arrows indicate co-localization of IF signals. Scale bar: 70 μm (upper panel) and 40 μm (lower panel).

D, E  Confocal images showing GALC-expressing astrocytes (S100β; D) and neurons (NeuN; E) in the contralateral hemispheres, anterior (slice 8) and posterior (slice 12) to the injection site. Arrows indicate co-localization of IF signals. Scale bars: 40 μm.

▶

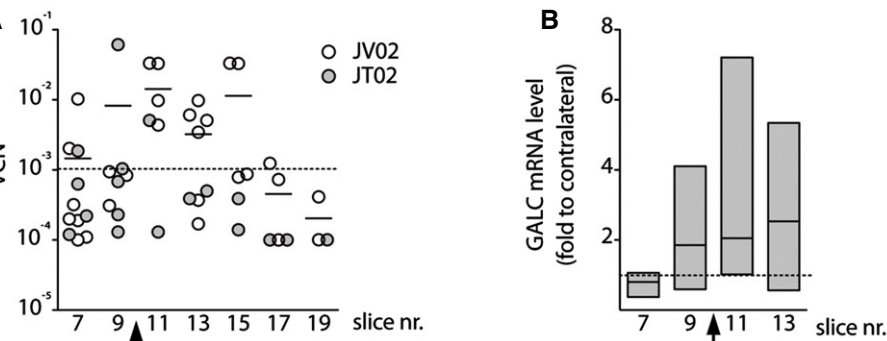

Figure 6.

**Figure 7. Gene therapy rescues GALC activity in CNS tissues of Krabbe-affected NHP.**

A Specific GALC enzymatic activity in single brain slices of WT untreated (UT; white bars, physiological levels), WT LV.hGALC-treated (JV02), and Krabbe-affected LV.hGALC-treated (JV02) animals (filled gray bars, injected hemisphere; striped gray bars, contralateral hemisphere). Two-way ANOVA followed by Bonferroni's multiple comparison tests; *$P < 0.05$, **$P < 0.01$ versus matched slices of WT UT.

B, C Summary of GALC activity in brain (B) and spinal cord (C; $n = 3$ blocks/animal) samples from JT02 and JV02, as well as from WT and Krabbe UT controls.

D Chromatographic profile of GALC in selected brain slices and spinal cord (SC) tissue of untreated (UT) WT and LV.hGALC-treated Krabbe NHP (JT02). Extracts were run through a Sephadex S-300 gel filtration column. Fractions (0.2 ml) were assayed for enzyme activity using MUGAL substrate in the presence (O) or absence of 11 μM AgNO$_3$ (●).

E GALC activity in the CSF of JT02 (pre- and post-treatment) and JV02 (post-treatment). Dots represent technical replicates.

F Beginning at 2 months (m) of age, animals were evaluated monthly for motor performance using a modified Bayley's scale for infant development. For the study animals (JV02 and JT02), scores are reported as pre-surgery and post-surgery. Affected animals score significantly lower than normal animals at any age considered. JT02's motor scores increase over time post-LV.GALC injection, being close to the mean observed in normal juveniles at 5 months. Historical data for normal and affected animals are presented as means plus one standard deviation. Historical data are analyzed by two-way ANOVA and Bonferroni's posttests, $P < 0.05$ at 6 months, $P < 0.001$ at all other ages in affected versus normal animals; $n = 5–20$ animals/group, except for normal animals at 6 months ($n = 1$). See also Appendix Table S9.

Data information: Data in (A–C) are represented as floating bars (min to max, line at mean; $n = 2–6$ blocks/slice). Number of blocks analyzed/animal: 16 (WT UT), 24 (JT02), and 22 (JV02). GALC activity in Krabbe UT samples is $< 0.001$ nmol/h*mg; $n = 3$ blocks. One-way ANOVA and Bonferroni's multiple comparison test (B) or Dunnet's multiple comparison test (C); *$P < 0.05$, **$P < 0.001$ versus WT UT.

provides stable therapeutically relevant enzyme activity in the whole brain, spinal cord, and liquor of treated animals after injections in only two CNS regions. A consistent pattern of vector and enzyme biodistribution was observed in LV.hGALC- and LV.hARSA-injected NHP even in the presence of differences in treatment protocols (i.e. diseased versus normal animals, different transgenes, vector dose, injected volume, and delivery system), suggesting the robustness of the proposed LV-mediated gene therapy platform and providing the first preliminary evidence of therapeutic efficacy in the exclusive NHP model.

By comparing results from the two experimental systems, it appears that vector dose, injection in posterior white matter regions, and CED-mediated delivery (Lonser *et al*, 2015), which allowed homogeneous and reproducible vector distribution, can be considered as factors that positively affect the extent of transduced volume and the enhancement of LV particle dispersion. The transduced brain volume in this study was overall lower when compared to that obtained following intracerebral AAV delivery (Colle *et al*, 2010; Rosenberg *et al*, 2014; Zerah *et al*, 2015). This can be explained by the multiple injection protocols (six injection sites with 6–12 deposits) used in those studies as compared to the two-injection protocol in this study. Also, it is in line with a more limited diffusion capacity of LV as compared to AAV, likely due to the larger size and, possibly, surface features of the lipid-bound LV particle. By delivering low vector doses ($0.35–1 \times 10^8$ TU/brain) in few intraparenchymal sites, we limited the invasiveness of the procedure while achieving efficient gene transfer in the targeted CNS regions, in which we documented transgene overexpression in neurons and glial cells. As previously shown in the murine CNS (Lattanzi *et al*, 2010, 2014), LV proficiently transduced NHP oligodendrocytes. This specific feature is important in the perspective of clinical development. Indeed, oligodendrocytes are the more abundant cell types in white matter areas and the most affected cell population in MLD and GLD but are hardly transduced by other vector types, including AAV (Colle *et al*, 2010; Piguet *et al*, 2012; Rosenberg *et al*, 2014). Also, while enzyme-deficient oligodendrocytes are efficiently cross-corrected *in vitro* or in the mouse brain (Lattanzi *et al*, 2010; Piguet *et al*, 2012; Santambrogio *et al*, 2012), this mechanism appears less efficient in large brains (Colle *et al*, 2010).

The transduction of a small pool of endogenous cells established a widespread expression of transgenic products. The presence of

close to normal (GALC) or supranormal (ARSA) enzymatic activity, even in the most caudal brain regions and in spinal cord at 3 months post-gene therapy (but likely even at earlier time points, based on levels of transgene expression observed 1 month post-GT in pilot animals) confirmed long-distance enzyme distribution by possible axonal transport (Ciron *et al*, 2009; Colle *et al*, 2010; Rosenberg *et al*, 2014) and cross-correction, the property of lysosomal enzymes to be secreted and recaptured by surrounding cells. Moreover, the presence of twofold normal ARSA activity and > 10% of normal GALC activity in the liquor of treated NHP (while no GALC protein could be isolated from the CSF of Krabbe-affected NHP before treatment) suggested sustained production and efficient bioavailability of the enzymes. Of note, the 100% increase of ARSA levels observed in the physiological background at 3 months post-treatment are comparable to the physiological ARSA levels detected in the liquor of MLD patients 12–24 months post-HSC gene therapy (Biffi *et al*, 2013). These results provide a rationale to propose this approach to stabilize CNS damage and prevent further deterioration in the late-onset forms of MLD and GLD but also to counteract the rapid progression of CNS pathology in early symptomatic infantile/early juvenile patients. While a long-term follow-up would be needed to assess persistency of enzymatic levels in our model, our results point against the occurrence of vector silencing or immune response against the vector or transgenes. Of note, antibodies against envelope and capsid proteins were occasionally and transiently seen in NHP and PD patients 3 and 6 months after administration of a lentiviral vector-based therapeutics delivering dopaminergic-related genes (ProSavin) without affecting transgene expression (Jarraya *et al*, 2009; Palfi *et al*, 2014).

Studies of healthy individuals carrying pseudodeficient variants of the *ARSA* gene suggest that 10% of normal ARSA activity is sufficient to prevent the onset of MLD symptoms (Gieselmann, 2006). Similarly, while all individuals with Krabbe disease have very low GALC enzyme activity (0–5% of physiological levels), there is a relatively broad range of GALC activities in the healthy population. Indeed, the presence of multiple polymorphisms in both *GALC* alleles might lower enzyme activity to 8–20% of normal without resulting in clinical disease (Wenger, 1993, 2000). In this view, the extent of ARSA and GALC expression and activity observed in the brain of LV.hARSA- and in the Krabbe-affected LV.hGALC-treated

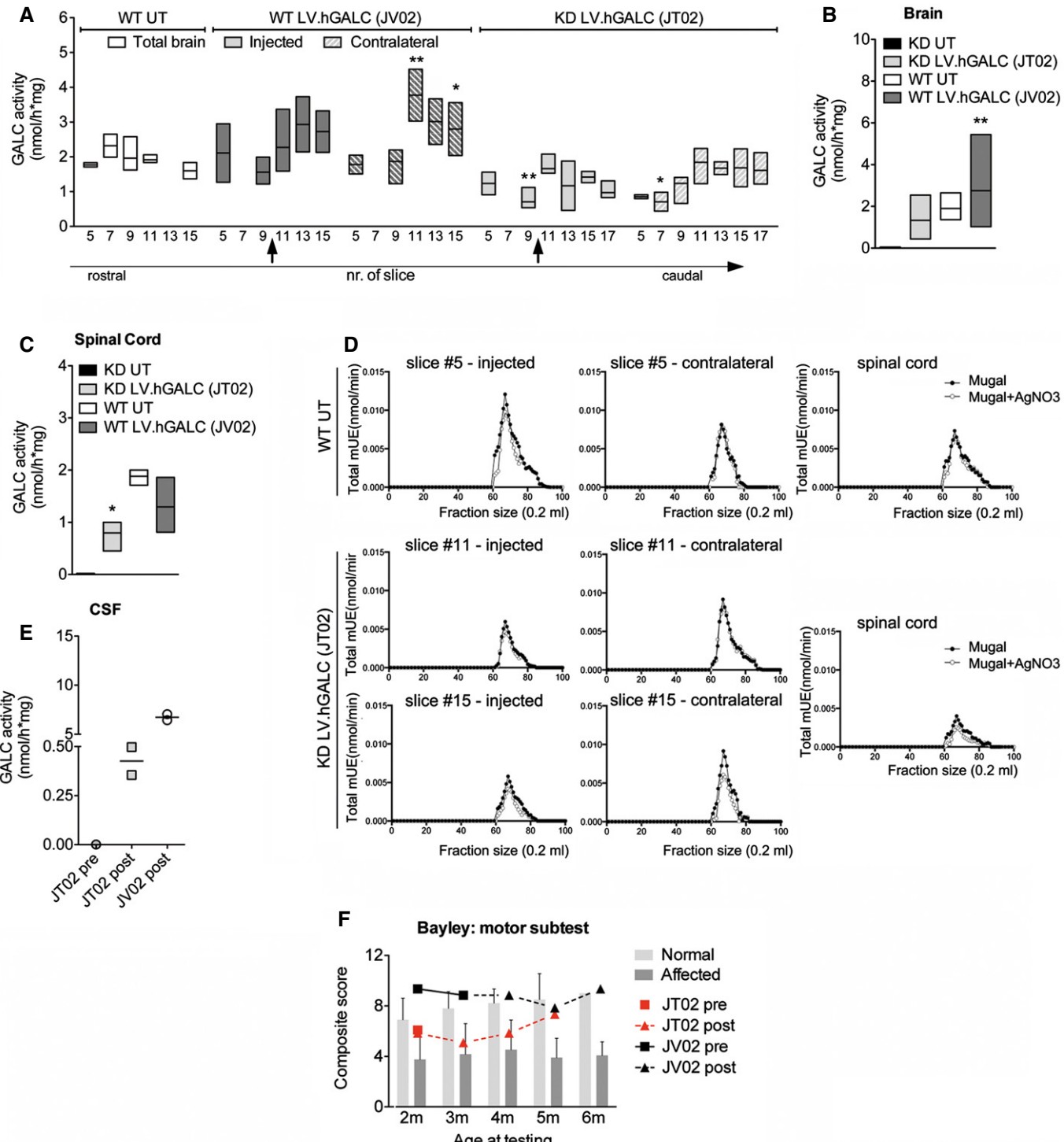

Figure 7.

NHP are compatible with foreseeable therapeutic activity in patients, which remains to be evaluated in rigorous clinical trials in which selective inclusion/exclusion criteria as well as biochemical end points will be key to correctly estimate the clinical impact of this approach and to highlight potential therapeutic hurdles that are dependent on the biochemical and kinetic properties of the individual enzymes.

The local neuroinflammation in the vicinity of the injection areas, which we found in some treated animals, was previously reported in AAV- (Colle et al, 2010; Salegio et al, 2012; Rosenberg et al, 2014) and LV-injected NHP (Kordower et al, 2000; Jarraya et al, 2009), being mainly associated with tissue damage (consequent to the surgical procedure), persistence of injected soluble material, and/or immune response. Our data suggest that macro-

phage and mononuclear cell infiltrates observed to a different extent in LV.GFP-, LV.hARSA-, and, indirectly, in LV.hGALC-injected NHP at 3 months post-treatment are not correlated to an immune response against transgenes or LV capsid proteins. In contrast to a recent study (Zerah *et al*, 2015), recruitment of B and T cells at sites of neuroinflammation was not associated with anti-hARSA antibodies (undetectable in LV.hARSA-injected NHP, as expected considering the 96% identity between human and *Macaca fascicularis* ARSA proteins) or anti-GFP antibodies (detectable at low titer in only one of the two LV.GFP-injected NHP, both showing similarly mild inflammation). Also, only one out of five NHP showing mononuclear cell infiltration developed detectable antibodies against the VSV envelope and LV capsid protein, and similar titers of antibodies were detected in one NHP in which cell infiltration was absent. The mechanism by which LV injections may have caused these focal lesions remains unclear. Importantly, there was no clinical impact of these histopathological findings, as documented by normal general safety, hematological, and behavioral parameters.

The use of self-inactivating long-term repeats in new generation LV reduces the potential for gene activation at the insertion sites (Montini *et al*, 2006). LV-delivered *ARSA* and *GALC* genes do not impart a proliferative advantage to neural cells (Neri *et al*, 2011). This, in addition to the targeting of postmitotic cells (Colin *et al*, 2009; Lattanzi *et al*, 2014) and the preferential integration into the body rather than promoters of active genes, minimizes the risk of oncogenesis as compared to earlier generation retroviral vectors. Instead, promoter insertion and enhancer-mediated activation of oncogenes are the preferred mechanisms of insertion-driven oncogenesis (Cesana *et al*, 2014). Our earlier studies have shown that LV-transduced cells undergo proliferation and apoptosis associated with physiological neonatal brain development. In contrast, we never observed hyperproliferation of transduced cells or formation of tumor masses in mice analyzed at PND40 and at 6 months post-neonatal injection (Lattanzi *et al*, 2014). Intragenic LV insertions may interfere with normal cell function, triggering the formation of aberrant transcripts that result in gain or loss of function mutations (Cesana *et al*, 2012; Moiani *et al*, 2012). However, this mechanism has not been associated with cell transformation in the follow-up of HSC gene therapy trials, in which no adverse genotoxic events have been shown to date (Cartier *et al*, 2009; Aiuti *et al*, 2013; Biffi *et al*, 2013). It is also possible that mutant proteins produced by aberrant transcripts acquire detrimental functions and cause cell death or functional impairments. The impact of this phenomenon in non-proliferating tissues, however, would be essentially limited to the cell(s) in which the damaging insertion has occurred, and it is difficult to predict/assess in this experimental setting. The polyclonal pattern of LV integration, the preferential targeting of neural-specific genes without preferential insertions in genes involved in tumor formation or neurological disorders that we report here for the first time in NHP are in line with previous data obtained in rodents (Bartholomae *et al*, 2011; Lattanzi *et al*, 2014) and further point to a low genotoxic risk associated with the proposed GT platform. The CNS-restricted distribution of LV particles (Jarraya *et al*, 2009; Lattanzi *et al*, 2014) and, consequently, the absence of off-target LV integration in the periphery represent an additional favorable safety trait of this GT platform.

According to previous studies performed in mice (brain volume $\approx 0.5\ cm^3$) (Lattanzi *et al*, 2010, 2014), we estimated that

$\approx 1 \times 10^6\ TU/cm^3$ could be safe and potentially effective in achieving enzyme correction in the brain of juvenile NHP (brain volume $\approx 60–65\ cm^3$). Based on the respective brain volumes of NHP and human infantile/adult patients ($\approx 500–1,000\ cm^3$), the equivalent vector dose to be injected in patients would be $\approx 5 \times 10^8–1 \times 10^9$ TU. This roughly corresponds to $\approx 0.5–1\ ml$ of high-titer clinical grade LV batches (Biffi *et al*, 2013), which could be easily divided into multiple intraparenchymal vector deposits in order to minimizing the vector dose at each injection site while maintaining robust cell transduction, transgene production, and expected biological efficacy.

We have previously demonstrated similar transduction efficiency and biodistribution of ARSA and GALC enzymes upon intracerebral LV gene delivery in mice. Here, we provide proof of concept that this equivalence holds true in larger brains. Considering the ethical and practical limitations of working with NHP, eight normal monkeys represent a considerable sample size to study safety and biodistribution in a physiological background. The rhesus monkey model of Krabbe disease represents the first reported observation of a lysosomal storage disease in any non-human primate species (Wenger, 2000). By treating one Krabbe-affected juvenile animal, we could assess for the first time ever vector and transgene biodistribution as well as rescue of enzymatic activity in CNS tissues of a large animal that shares high degree of pathological and clinical similarity to the human disease. This analysis provided formal proof of biological efficacy of the strategy, as it was predicted from the results obtained in LV.hARSA-injected normal NHP. Neurobehavioral assessment using scales standardized for use with juvenile rhesus macaques showed progressive improvement of motor performance (muscle tone and strength of muscle contractions) in the gene therapy-treated Krabbe NHP, which reached those of age-matched normal animals at 3 months post-gene therapy. Notably, motor scores of Krabbe-affected macaques generally fell below normal animals at any age considered and tend to decline with age, resembling the relentless motor deterioration observed in individuals with the infantile form of Krabbe disease (Wenger, 1993). So, our results provide indication (although very preliminary) of therapeutic benefit of the lentiviral gene therapy strategy in this model. Given the variability in disease progression that characterize Krabbe-affected macaques (Baskin *et al*, 1998; Weimer *et al*, 2005; Borda *et al*, 2008), further studies including more animals might better address potential immune response against the GALC protein (a minor issue if considering patients with residual protein and enzymatic activity) and definitely prove whether metabolic rescue translates into clinical–pathological amelioration and overall therapeutic benefit.

In conclusion, our study advances intracerebral lentiviral gene therapy for clinical development. Lentiviral vectors may cooperate with other gene/cell therapy strategies to address some of the unresolved challenges that have limited so far the development of effective treatment options for MLD and GLD patients.

# Materials and Methods

### Study design

Non-human primate studies were limited in sample size for feasibility and ethical reasons. For experiments in normal NHP, we chose to

use *n* = 2–3 animals/treatment group (random assignment) in order to reliably detect potential differences related to treatment. In contrast, only one Krabbe-affected monkey and one normal monkey of comparable age were available for treatment with LV.hGALC. However, the use of a similar experimental protocol in the two studies (age at injection, injection sites, tissue collection and analysis, and readouts) and the availability of historical data/samples from the Krabbe NHP colony allowed us to go beyond a qualitative assessment of data. The sample size of mouse studies was chosen according to earlier studies of intracerebral LV gene therapy (Lattanzi *et al*, 2010, 2014). Mice were randomly assigned to experimental groups. Untreated mice (mutant and WT) served as controls for the treatment groups. No animal (mice and NHP) administered the intended dose and surviving the procedures was excluded from the analysis.

Whenever possible results are shown in dot-plot graphs or in low–high bar graphs (line at mean) in order to show intragroup variability. Investigators involved in the histopathological analysis were blinded. Investigators performing animal handling, sampling, euthanasia, and raw data analysis were not blinded.

## Non-human primates

### Normal NHP

Ten pregnant females (*Macaca fascicularis)* were purchased by BIOPrim, Baziege, France, and housed at MIRCen, CEA/INSERM, Fontenay aux roses, France. Nine babies (two females and seven males) were delivered and were used for this study when they reached 2–3 months of age (juveniles). Details on the animals used can be found in Appendix Table S1.

### Krabbe-affected NHP

One 53-day-old Krabbe-affected rhesus monkey (*Macaca mulatta)* and one age-matched unaffected normal animal from the colony housed at TNPRC were used. Tissues from four Krabbe-affected monkeys (available in Prof. Bunnell's lab) were used as controls for molecular and enzymatic analyses. Details on the animals used can be found in Appendix Table S7.

## Plasmids and vector production

Low-endotoxin plasmid containing the full-length coding sequence of the human (h)ARSA and VSV-pseudotyped lentiviral vector (LV) batches was prepared by MolMed S.p.A. (http://www.molmed.com) and manufactured according to the process approved for clinical use (Biffi *et al*, 2013) and thus subject to stringent quality assessment, although the procedures were not performed under GMP in order to contain costs.

The plasmid containing the full-length coding sequence of the human (h)GALC gene was kindly provided by Dr. Shen, Baylor Univ. Medical Center, Dallas, USA. The human gene coding for GALC was tagged with the Myc epitope. Our *in vitro* and *in vivo* data on murine models indicate that Myc epitope does not interfere with the lysosomal targeting and enzymatic activity.

Laboratory grade/scale VSV-pseudotyped third-generation LV encoding for green fluorescent protein (LV.GFP) or hGALC-myc (LV.hGALC) under the control of the human phosphoglycerate kinase (PGK) promoter was produced by transient four-plasmid co-transfection into 293T cells and purified by ultracentrifugation,

according to established protocols (Amendola *et al*, 2005). Reagents, cloning procedures, and sequence information are available upon request. Infection titer was estimated on 293T cells by limiting dilution (LV.GFP) or by qPCR (LV.hGALC). Vector particle was measured by HIV-1 gag p24 antigen immunocapture (NEN Life Science Products, Zaventem, Belgium). Vector infectivity was calculated as the ratio between titer and particles. Details on LV batches used can be found in Appendix Table S2. Eight vials (1 ml/vial) of LV.hARSA, LV.GFP (100 μl/vial), and LV.hGALC (100 μl/vial) were randomly selected, stored frozen, and delivered to MirCen (LV.hARSA and LV.GFP) and to TNPRC (LV.hGALC) 2 weeks before the scheduled day of treatment.

## Surgical procedures

### LV injection using convection enhanced delivery (CED)

After underskin epinephrine–xylocaine local administration, animals were positioned in a MRI-compatible stereotaxic apparatus. Unilateral (left hemisphere) craniotomy was performed to position a CED cannula to infuse LV.GFP or LV.hARSA ($2 \times 10^7 – 5 \times 10^7$ TU/injection site) into two targeted regions of the brain parenchyma, namely anterior external capsule/corona radiata and thalamus (pilot group and study group 1); anterior external capsule/corona radiata and posterior external capsule (study group 2). Surgical coordinates were determined from preoperative MRI using the known distance from the scan plane containing the ear bars (anteroposterior axis) and the mediolateral distance from midline on the coronal image containing the target. The infusion system consisted of a fused silica reflux-resistant cannula connected to a loading line (containing vector suspension) and an infusion line (containing sterile saline). A 1-ml syringe mounted on an MRI-compatible infusion pump (Harvard Apparatus Inc, Holliston, Massachusetts, USA) regulated the flow of fluid through the delivery cannula. Approximately 40–150 μl of vector suspension was infused through the cannula via infusion pump to the target site with a dosing scheme consisting of ascending infusion rates. Immediately after the infusion procedure, the wound site was closed in anatomical layers, and the animal was transferred to the MRI room for a post-infusion scan. After that, the animal was returned to the operating room for extubation and recovery procedures.

### LV injection in Krabbe-affected (JTO2) and normal (JVO2) NHP

Animals were anesthetized with ketamine hydrochloride 10 mg/kg IM, intubated, and maintained on isoflurane and oxygen. A midline skin incision was made over the dorsal aspect of the skull. All muscle and subcutaneous tissue were reflected with a periosteal elevator and the skin and underlying tissue is retracted to ensure constant exposure of the skull. The animals were positioned in a MRI-compatible stereotaxic frame (1430M, Kopf Instruments). The stereotaxic instrument was mounted on the stereotaxic head frame and rostral/caudal and medial/lateral coordinates (achieved with a Micromanipulator; model 1760-61 Kopf instruments) were located just above the skull. The stereotaxic instrument was removed to allow for drilling an access hole into the skull. A 4-mm drill bit was used to drill through the outer cortex and medullary cavity of the skull. A constant sterile water lavage of the drilling site was performed to minimize airborne bone dust and to keep surrounding tissues cool. A 1.5-mm drill bit was used to drill through the inner

cortex and clean the margins of the defect to ensure that no damage occurs to underlying structures. Once adequate exposure to specific anatomical locations of the brain is obtained, the stereotaxic instrument was placed back on the stereotaxic head frame. The predetermined rostral/caudal and medial/lateral coordinates of the brain were obtained by adjusting the stereotaxic instrument. Injections were performed with a 50-μl Hamilton syringe with a 2-inch 22-gauge needle. Injection rate was approximately 1 μl/min. Coordinates for JV02: injections were performed in the right hemisphere, in a rostrocaudal plane at the level of the external auditory meatus. One injection was performed in the thalamus (15 mm ventral to the surface of the brain and 3 mm lateral to midline; $2 \times 10^7$ TU/40 μl), and three injections were performed in the internal capsule (15.5 mm ventral, 8.5 mm lateral; 13 mm ventral, 8 mm lateral; 10 mm ventral, 6.5 mm lateral; $5 \times 10^6$ TU/20 μl/site). Coordinates for JT02: injections were performed in the right hemisphere, in a rostrocaudal plane 4 mm cranial to the external auditory meatus. One injection was performed in the thalamus (19 mm ventral and 3 mm lateral; $2 \times 10^7$ TU/40 μl), and three injections were performed in the internal capsule (23 mm ventral, 7 mm lateral; 19 mm ventral, 7 mm lateral; 19 mm ventral, 9 mm lateral; $5 \times 10^6$ TU/20 μl/site). Total TU injected/brain for JT02 and JV02 was $3.5 \times 10^7$.

**Quantification of VCN**

VCN in murine samples was quantified as previously described (Lattanzi *et al*, 2010). Genomic DNA was extracted from NHP brain tissues, spinal cord, sciatic nerve, liver, spleen, and gonads following manufacturer's instructions (#200600, DNA Extraction Kit, Agilent Technologies, Inc. Santa Clara, CA 95051, USA). Samples were quantified by NanoDrop ND-1000 Spectrophotometer (Euroclone, Pero, Italy). Vector copies per genome were quantified by TaqMan analysis using 50 ng of template DNA. DNA extracted from untreated NHP was used as negative control. LV backbone was amplified by using forward primer (5′-TACTGACGCTCTCGCACC-3′ – final concentration: 10 μM), reverse primer (5′-TCTCGACGCAG GACTCG-3′ – final concentration: 10 μM), and FAM/MGB probe (5′-FAM-ATCTCTCTCCTTCTAGCCTC-MGB-3′ – 5 μM final concentration). As there was an internal reference gene for normalization, we amplified a fragment of the *Macaca fascicularis* TAF7 RNA polymerase II gene (TATA box-binding protein-associated factor) by using a Custom TaqMan Gene Expression Assays containing both probe and primers (Rh02916247_s1, reporter: FAM, quencher: MGB, Life Technologies). Other details can be found in Appendix Supplementary Methods.

LV diffusion along the three axes was calculated considering consecutive blocks with detectable VCN along the axis. The volume of each brain block was estimated by dividing the average hemisphere volume of NHP in the study (32.5 cm³; estimated through MRI analysis and consistent with published volumes for the *Macaca fascicularis* and *Macaca mulatta* brain) by the number of blocks per hemisphere. P1 NHP: 0.25 cm³; P2 NHP: 0.37 cm³; Study groups: 0.32 cm³. JT02 and JV02: 0.25 cm³.

**Quantification of ARSA and GALC mRNA**

Blocks with detectable VCN as well as surrounding blocks in the same or different slices (injected hemisphere) and matched brain blocks of the contralateral hemisphere were selected for mRNA extraction. Total RNA was isolated and purified from brain tissues using RNeasy Lipid Tissue Kit (Qiagen, Hilden, Germany), following the manufacturer's instructions. The quantity of RNA was determined by 260/280 nm optical density reading on NanoDrop ND-1000 Spectrophotometer (NanoDrop, Pero, Italy). Reverse transcription was carried out using 2 μg of total RNA and the Quantitect Reverse Transcriptase kit (Qiagen, Hilden, Germany).

We evaluated the exogenous hARSA and hGALC mRNA expression by quantitative PCR, using primers (forward, 5′-GGCTGT TGGGCACTGACAAT-3′, final concentration: 10 μM; reverse, 5′-ACG TCCCGCGCAGAATC-3′, final concentration: 10 μM) and probe (5′-FAM-TTTCCTTGGCTGCTCGCCTGTGT-MGB-3′, final concentration: 5 μM) annealing the vector construct WPRE sequence. mRNA samples extracted from brain of UT NHP were used as negative controls.

In parallel, we assessed total ARSA and GALC mRNA expression using a Custom TaqMan Gene Expression Assays, containing both probe and primers (ARSA: Rh02828799_m1 reporter: FAM, quencher: MGB, Life Technologies; GALC: Rh02827142_m1 reporter: FAM, quencher: MGB, Life Technologies). As an internal reference for normalization, we amplified a fragment of the *Macaca fascicularis* TAF7 RNA polymerase II gene, using a Custom TaqMan® Gene Expression Assays, containing both probe and primers (Rh02916247_s1, reporter: FAM, quencher: MGB, Life Technologies). Reactions were carried out in a total volume of 12.5 μl, in a ViiA™ 7 Real-Time PCR System (Life Technologies-Applied Biosystems, Carlsbad, CA, USA).

Expression levels of transgenic hARSA and hGALC mRNA were calculated as fold to TAF7 expression levels. Samples with WPRE probe CT > 37 (corresponding to a fold value < 0.0001) were considered as having undetectable levels of exogenous hARSA or hGALC expression.

Total ARSA and GALC mRNA expression levels in tissue blocks of the injected hemisphere were expressed as fold to mRNA expression levels detected in pair-matched blocks of the contralateral hemisphere. Only samples with TAF7 probe CT within 22–26 were considered in the analyses. Data were obtained by $n = 2$ experiments performed in duplicate. Within the same experiments, samples with a CT variability > 2 for TAF7 probe CT, WPRE probe CT, total GALC probe CT, or total ARSA probe CT within the duplicate were not considered in the analyses. Samples with interexperiment variability > 50% were analyzed at least in $n = 3$ experiments.

**Immunofluorescence**

Selected post-fixed samples from brain were recovered from paraffin using xylene and then rehydrated with different dilutions of ethanol in water. Unmasking was performed in citrate/Tris–EDTA buffer in a bain-marie for 30 min. Slides were incubated with blocking solution [0.5% milk + 10% normal goat serum (NGS) + 1% BSA + 0.3% Triton X-100 in PBS] for 2 h at RT and then incubated overnight at 4°C with primary antibody diluted in blocking solution. After washing in PBS (3 × 10 min each), antibody staining was revealed using species-specific fluorophore-conjugated secondary antibodies diluted in 10% NGS + 1% BSA + 0.1% Triton X-100 in PBS. Tissue sections were counterstained with DAPI (4′,6-diamidino-2-phenylindole; 10236276001,

Roche, Basel, Switzerland) or ToPro-3 (T3605, Life Technologies, Invitrogen, Carlsbad, CA, USA), washed in PBS and mounted on glass slides using Fluorsave (#345789, Calbiochem, Billerica, MA, USA).

## Cell counts

GFP staining was evaluated in coronal brain sections selected close to the anterior or posterior injection sites (two blocks, two slices/block, and five fields/slice). GFP staining was performed in combination with lineage-specific markers (NeuN, neurons; GFAP, astroglia; APC and CNPase, oligodendroglia) and nuclear counterstaining. Triple-labeled sections were scanned with Perkin Elmer UltraVIEW ERS spinning disk confocal microscope at 40× magnification; GFP$^+$ cells showing defined nuclei were counted and co-localization with neuronal or glia markers was evaluated. Results were expressed as follows: (i) percentage of GFP$^+$ cells on total number of nuclei (extent of transduction); (ii) percentage of double-positive GFP$^+$NeuN$^+$, GFP$^+$GFAP$^+$, or GFP$^+$CNPase$^+$ cells over the total number of GFP$^+$ cells.

## Immunohistochemistry

Post-fixed tissues samples were stained with: (i) anti-CD3, anti-CD20, and anti-CD11c to assess the nature of infiltrating cells. Detection of signal was performed with UltraVision Quanto Detection System HRP DAB (Thermo Scientific™); (ii) anti-ARSA antibody recognizing an epitope conserved in both human and *Macaca fascicularis* by using Ventana autostainer. Samples included cortical and subcortical white and gray matter regions of the left and right hemispheres of slices 3–6. All specimens were examined in blind. ARSA-positive signal was quantified, and a score was assigned based on the following factors: pattern of distribution of stained cells, approximate intracellular granulation size and number (low versus high quantity of granules), and the level of the staining in the surrounding extracellular matrix (ECM). The sum of the partial scores gave the final grade for each specimen. For each NHP, the mean score was calculated and data were expressed as fold increase to the score assigned to P2 NHP (control).

## Enzymatic activity and DAE chromatography

ARSA and GALC activity assay as well as DAE chromatography were performed according to previously described protocols (Martino *et al*, 2005, 2009; Morena *et al*, 2014), which have been validated in our previous studies (Lattanzi *et al*, 2010, 2014; Neri *et al*, 2011).

## Detection of antibodies in the serum (ELISA assay)

Pre- and post-surgery sera collected from LV-injected NHP were tested to detect the presence of antibodies against transgenes (GFP and hARSA) and LV particles through enzyme-linked immunosorbent assay (ELISA). Details can be found in Appendix Supplementary Methods.

## p24 enzyme-linked immunosorbent assay

The presence of vector particle in the sera of LV-injected NHP was determined by immunosorbent assay by using the Alliance HIV-1

p24 antigen ELISA kit (Perkin Elmer), following manufacturer's instructions. Details can be found in Appendix Supplementary Methods.

## LAM-PCR and 454 pyrosequencing

The procedures and primer sequences for linear amplification-mediated (LAM)-PCR for LV integration site retrieval have been previously described (Schmidt *et al*, 2007). Briefly, we used 1 μg of DNA extracted from brain tissues as template for 100-cycle linear PCR pre-amplification of genome–vector junctions, followed by magnetic capture of the biotinylated target DNA, hexanucleotide priming, restriction digest using Tsp509I and HpyCHIV4, and ligation of a restriction site–complementary linker cassette. The first exponential biotinylated PCR product was captured via magnetic beads and re-amplified by a nested second exponential PCR. We separated LAM-PCR amplicons on high-resolution Spreadex gels (Elchrom Scientific) to evaluate PCR efficiency and the bands pattern for each sample. We adapted the LAM-PCR samples for 454-pyrosequencing by fusion primer PCR to add the Roche 454 GS-FLX adaptors (Paruzynski *et al*, 2010): Adaptor A, plus a six nucleotides barcode, was added at the LTR end of the LAM-PCR amplicon; adaptor B was added at the linker cassette side. Each sample was amplified with fusion primers carrying each a different barcode sequence. Fusion primer PCR products were assembled in equimolar ratio into libraries, avoiding repetition of identical barcodes, and sequenced at GATC biotech. Sequence reads were mapped to reference genome of *Macaca Fascicularis*, with a web-available version of BLAST. All the hits with an identity score lower than 95% were discarded, as well as those with a starting alignment position beyond the third base. Each integration site (IS) was annotated using an own developed annotation tool (Calabria *et al*, 2014). For each IS, the output of the tool contains the following information: chromosome position, feature's name (in which the IS is contained), starting and ending position, distance of the IS from the feature's transcription starting site, and the IS relative position with respect to the feature. Raw sequence reads are available at the NCBI sequence reads archives (SRA) at the following link: http://www.ncbi.nlm.nih.gov/bioproject/PRJNA311501.

The raw data were converted to a list of GeneID, using DAVID online software (http://david.abcc.ncifcrf.gov/home.jsp) and then filtered using the same software. Gene Ontology was performed using the functional annotation tool. Results were filtered with the following parameters: *P*-value < 0.05; fold change > 2; Benjamini value < 0.05; false discovery rate < 0.05.

## Statistics

Data were analyzed with GraphPad Prism version 5.0a for Macintosh and expressed as the mean ± standard error mean (SEM) if not otherwise stated. Nonparametric (Kruskal–Wallis) or parametric tests (Student's *t*-test, and one-way or two-way analysis of variance) followed by *ad hoc* post-tests were used according to data sets. The number of samples analyzed and the statistical test used are indicated in the legends to each figure. The summary of statistics for the main figures and tables is provided in Appendix Table S10.

## Study approval

Mouse studies were conducted under an approved protocol of the "Institutional Committee for the Good Animal Experimentation" of the San Raffaele Scientific Institute and are reported to the Ministry of Health, as required by Italian law.

All the procedures and protocols involving NHP were performed under approval of: (i) MIRCen ethical committee, in accordance with the animal welfare guidelines and recommendations provided by the directive 2010/63/UE (decree 2013-118) (normal NHP); (ii) Institutional Animal Care and Use Committee (IACUC) of the TNPRC conformed to the requirements of the Animal Welfare Act (Normal and Krabbe-affected NHP).

For details on mouse studies, *in vitro* studies, surgical and post-surgical monitoring, magnetic resonance imaging, tissue collection and processing, quantification of VCN, Western blot analyses, histopathology, ELISA assay, p24 enzyme-linked immunosorbent assay, and primary and secondary antibodies, see the Appendix Supplementary Methods.

**Expanded View** for this article is available online.

## Acknowledgements

We thank Lucia Sergi Sergi for the help with LV.GFP and LV.hGALC production; Tiziana Plati for the help with p24 assay; Fabrizio Benedicenti for the help with LAM-PCR; Andrea Annoni, Alessio Cantore, and Silvia Gregori for the help with immune response studies and critical discussion of data; MolMed S.p.A for LV.hARSA production; Alessandra Biffi for critical discussion of data; Eleonora Calzoni for analysis of ARSA activity using sulfatide; the MirCen team: Lev Stimmer and Helene Touin for histopathology, Laurent Watroba for coordinating surgery procedures, Julien Flament for MRI analysis and 3D brain reconstruction, Francois Lachapelle for coordinating the MiRCen staff dedicated to the project, all the vets and technicians providing professional support during the duration of the project; and the TNPRC team: Peter J. Didier for necroscopy and histopathology and Cyndi Trygg for technical support and logistics. This work was supported by: Fondazione Telethon, grant number TGT11B02 to AG; NIH grant OD 11104 to the TNRPC; TNPRC Pilot Research project to BAB and AG.

## Author contributions

VM designed and performed experiments, analyzed the data, and wrote the manuscript. AL produced LV.hGALC and performed mouse studies. GB performed molecular analyses, Western blot, immune response studies, and cytokine expression profile. LT performed immunofluorescence analysis on LV.GFP-injected NHP and molecular analysis on LV.hGALC-injected NHP. KB contributed to the experimental plan for the studies performed at MirCen and performed surgery using CED-mediated delivery. JB coordinated logistics and helped with CED-mediated delivery and data collection. FM performed ARSA and GALC activity assays. SM supervised biochemical assays, analyzed the data, and critically revised the manuscript. AC performed integration site analyses and analyzed the data. FS and CD performed immunohistochemistry on NHP tissues. EM supervised integration site studies and analyzed the data. JPD performed surgery at TNPRC. JMFP performed immunofluorescence on tissues from NHP injected at TNPRC. KCB supervised the behavioral analysis at TNPRC; BAB coordinated the studies performed at TNPRC. LN contributed to the experimental plan and critically discussed data. AG designed the experimental plan, supervised the project and facilitated collaboration between the different parties, analyzed the data, and wrote the manuscript.

### The paper explained

#### Problem

Metachromatic leukodystrophy (MLD) and globoid cell leukodystrophy (GLD or Krabbe disease) are recessive lysosomal storage diseases (LSD) caused by arylsulfatase A (ARSA) and galactosylceramidase (GALC) deficiency, respectively, leading to severe demyelination, neurodegeneration and premature death, and currently lacking any approved treatment. A major goal in the design of novel treatments is to achieve stable and widespread distribution of the therapeutic enzymes across the blood–brain barrier in CNS tissues. This rationale was recently validated in a clinical trial of hematopoietic stem cell (HSC)-based gene therapy (GT) for MLD (Biffi *et al*, 2013). However, the enzymatic reconstitution of CNS tissues mediated by HSC GT takes a few months to establish and might not benefit the fast-developing disease in the majority of symptomatic patients. So, there is a strong rationale for the development of alternative/complementary strategies able to provide rapid and efficient supply of the functional enzyme in CNS tissues, in order to counteract disease progression and/or stabilize the pathology.

#### Results

After many years of pre-clinical studies in MLD and GLD murine models, we tested here the safety and efficacy of LV-mediated GT in a considerable number of normal juvenile non-human primates (NHP) and in the unique NHP model of Krabbe disease, the only described LSD model in any non-human primate species. Our data provide strong evidence of biological efficacy and preliminary indication of therapeutic benefit. We proved that a small pool of neural cells transduced at high efficiency by low LV doses stably express and release transgenic enzymes that are functionally and biochemically indistinguishable from the native enzymes, at levels that can be foreseen as therapeutic if achieved in human patients. We showed for the first time undetectable genotoxicity related to integration of LV genome in neural cells in the context of intracerebral GT in juvenile NHP. These results, coupled to the absence of off-target LV integration in visceral organs, and to negligible immune response against LV and transgenes, represent additional favorable safety traits of this GT platform.

#### Impact

The results of the comprehensive study performed here in normal juvenile NHP using a clinically relevant vector should greatly facilitate the clinical development of this strategy. Importantly, by treating one Krabbe-affected juvenile animal, we assessed for the first time vector and transgene biodistribution as well as rescue of enzymatic activity in CNS tissues of a large animal that share high degree of pathological and clinical similarity to the human disease. These results suggest that the LV GT platform might be used to counteract the rapid progression of CNS pathology in early symptomatic infantile/early juvenile patients, but also to stabilize CNS damage and prevent further deterioration in the late-onset forms of the diseases. The broad distribution of enzymes throughout the nervous system and the cross-correction of endogenous non-transduced cells (enzyme-deficient cells, in the case of Krabbe-affected NHP) extend the potential impact of this strategy in the field of LSD and, likely, other neurodegenerative diseases.

## Conflict of interest

LN is an inventor on pending and issued patents on lentiviral vector technology filed by the Salk Institute, Cell Genesys, Telethon Foundation and/or San Raffaele Scientific Institute. The other authors declare that they have no conflict of interest.

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
