## [Review Process File · EMBO Molecular Medicine]

Pervasive supply of therapeutic lysosomal enzymes in the CNS of normal and Krabbe-affected non-human primates by intracerebral lentiviral gene therapy

Vasco Meneghini, Annalisa Lattanzi, Luigi Tiradani, Gabriele Bravo, Francesco Morena, Francesca Sanvito, Andrea Calabria, John Bringas, Jeanne M. Fisher-Perkins, Jason P. Dufour, Kate C. Baker, Claudio Doglioni, Eugenio Montini, Bruce A. Bunnell, Krystof Bankiewicz, Sabata Martino, Luigi Naldini and Angela Gritti

Corresponding author: Angela Gritti, IRCCS San Raffaele Scientific Institute

Review timeline:

Submission date:	17 September 2015
Editorial Decision:	23 October 2016
Revision received:	29 December 2005
Editorial Decision:	05 February 2016
Revision received:	22 February 2016
Accepted:	01 March 2016

Transaction Report:

Editor: Roberto Buccione

1st Editorial Decision

23 October 2016

Thank you for the submission of your manuscript to EMBO Molecular Medicine. We have now heard back from the three Reviewers whom we asked to evaluate your manuscript.

We are sorry that it has taken longer than usual to get back to you on your manuscript. In this case we experienced difficulties in securing appropriate reviewers and then obtaining their evaluations in a timely manner. Further to this, I wished to discuss the evaluations further with my colleagues

As you will see, based on the three evaluations from very expert reviewers, and especially those from reviewers 2 and 3, there are interconnected and major issues. Although I will not dwell into much detail, I would like to highlight the main points.

The first is that there appears to be a clear disconnect between two essentially different stories on different animal settings and with different reagents. Secondly, the possibly most exciting message of the manuscript - beyond the perhaps useful but a bit specialist technological development

described in the first part- is the clinically relevant treatment of the Krabbe NHP with LV-GALC, which however, is not considered to be sufficiently developed.

There are of course many other issues mentioned by the reviewers including some degree of overstatement given the lack of longer term follow-up, but I see the above as the major ones compromising suitability for publication.

Clearly, substantial re-writing and re-focusing and significant shortening are fundamental, and many clarifications required. The crux of the matter however, is the required experimentation (with respect to consolidating the Krabbe disease therapeutic approach) and its feasibility. On the other hand, perhaps you might have biological material and data available on which you could expand. will have to be toned down, given the limited experimental support. Without considerable improvements, I would find it difficult to support publication."

In conclusion, while publication of the paper cannot be considered at this stage, given the potential interest of your findings and after internal discussion, we have decided to give you the opportunity to address the above concerns.

We are thus prepared to consider a substantially revised submission, with the understanding that the Reviewers' concerns must be addressed and that acceptance of the manuscript will entail a second round of review. The overall aim is to significantly upgrade the relevance and usefulness of the dataset, which of course is of paramount importance for our title.

I understand that if you do not have the required data available at least in part, to address the above, this might entail a significant amount of time, additional work and experimentation and might be technically challenging, I would therefore understand if you chose to rather seek publication elsewhere at this stage. Should you do so, we would welcome a message to this effect.

Please note that it is EMBO Molecular Medicine policy to allow a single round of revision only and that, therefore, acceptance or rejection of the manuscript will depend on the completeness of your responses included in the next, final version of the manuscript.

As you know, EMBO Molecular Medicine has a "scooping protection" policy, whereby similar findings that are published by others during review or revision are not a criterion for rejection. However, I do ask you to get in touch with us after three months if you have not completed your revision, to update us on the status. Please also contact us as soon as possible if similar work is published elsewhere.

Please note that EMBO Molecular Medicine now requires a complete author checklist (<http://embomolmed.embopress.org/authorguide#editorial3>) to be submitted with all revised manuscripts. Provision of the author checklist is mandatory at revision stage; The checklist is designed to enhance and standardize reporting of key information in research papers and to support reanalysis and repetition of experiments by the community. The list covers key information for figure panels and captions and focuses on statistics, the reporting of reagents, animal models and human subject-derived data, as well as guidance to optimise data accessibility.

I look forward to seeing a revised form of your manuscript as soon as possible.

***** Reviewer's comments *****

Referee #1 (Remarks):

In the present manuscript, Meneghini and collaborators performed a pre-clinical study in non-human primates to assess the biodistribution, transgene expression, immune/inflammatory response, safety parameters and a pilot study on potential therapeutic benefit in one Krabbe-affected animal.

Overall this paper provides a strong basis for the potential use of LV for intracerebral administration of ARSA and GALC. The methods are appropriate and well described, the conclusions are balanced and adequately supported by the data. In the discussion, some of the limitations of the present study are clearly mentioned and discussed.

- 1) AAV-ASRA for the treatment of MLD patients is currently ongoing, it is unclear whether the present experimental setting will provide similar/additional therapeutic benefits?
- 2) What are the indications that MRI scan is a suitable method for the evaluation of the transduced area? Was it correlated with post-mortem analysis?
- 3) The authors mentioned that LV are transducing oligodendrocytes but the importance of this specific feature is not discussed in details.
- 4) Based on the NHP data it is not really possible to determine the added value of CED for LV delivery. Has a side-by-side comparison of the two delivery systems been performed?
- 5) Figure 3: the information provided by the pictures is extremely limited. High magnification pictures of the transduced areas and ARSA expression should be provided as supplemental Figures.
- 6) The term "infants" is used (page 16) to describe the study in Krabbe-affected monkey: Juvenile primate would be more appropriate.

Referee #2 (Comments on Novelty/Model System):

The study is based on the experimentation of intracerebral injection of lentiviral vectors expressing therapeutic enzymes in NHP models, one of which is a naturally occurring model of the lysosomal storage disorder Globoid Cell Leukodystrophy or GLC. These studies are meant to serve as preclinical assessment of the efficacy of this therapeutic approach for the treatment of this devastating disease in affected children.

Referee #2 (Remarks):

This paper gives a detailed account of an *in vivo* gene therapy study targeting primarily the CNS and performed in non-human primates (NHP) models. The study was designed to assess the therapeutic potential of two lentiviral (LV) vectors expressing the lysosomal enzymes arylsulfatase A (ARSA) and galactocerebrosidase (GALC) for the treatment of two severe neurodegenerative, demyelinating, lysosomal storage diseases (LSDs), MLD and GLC, respectively.

In recent years considerable progress has been made towards the development of therapeutics intended to correct or at least ameliorate the CNS disease in LSDs, and several clinical trials have started. However, effective treatment of the aforementioned and other neurodegenerative LSDs continue to be a big challenge for investigators and clinicians alike, and even in cases of ongoing clinical trials long-term patients follow-up are not yet available. In this respect, the current paper could be a valid contribution for both the scientific and clinical community, being the first report on LV-mediated intracerebral gene delivery in an NHP model of a lysosomal disease. Throughout the

experimental work has been conducted meticulously and accurately, including the necessary controls. The authors assessed volume and site of injection of the virus at two specific brain locations, chosen to allow for maximal spread of the viral particles; post LV injection, they determined the VCN, the levels of transgene RNA and protein expression across the injected and contralateral brain hemispheres, the spinal cord, the sciatic nerve and peripheral organs. They also verified the number of viral integration sites in brain tissue isolated from the injection site. Overall, the analyses are sound and carefully done.

What in my opinion is difficult to reconcile is the choice of NHP used in this study and the order in which the two main objectives (therapeutic use of LV-ARSA and LV-GALC) of the study are presented.

This taking into consideration that this group of investigators has previously contributed several studies evaluating different therapeutic strategies in models of MLD and GLC, including combination therapy consisting of intracerebral injection of LV-GALC vector and BMT in the Twitcher mouse model of GLC. The same group has also reported on the outcome of a clinical trial conducted in two MLD patients, one pre-symptomatic and the other at the onset of the symptoms. In this comprehensive study they described the beneficial effects of transplanting LV-modified autologous HSCs. Interestingly, they measured high levels of ARSA in the CSF of the treated patients and persistent expression of the therapeutic enzyme in hematopoietic cells with no evidence of clonal expansion of transduced cells. Very impressive results that will become a total break through if the patients will continue to do well in the long run.

Now, as rationale for embarking in the current study the authors argue that ex vivo BM gene therapy might not be sufficiently beneficial for post symptomatic patients given the lag of time potentially needed for CNS repopulation of BM derived cells carrying the therapeutic enzyme. This in spite of the high activity of enzyme measured in the CSF after HSCT, which should be indicative of high levels of circulating enzyme in the brain suited for cross correction of affected neural cells. However, the elaborate pilot studies based on the use of LV-ARSA in normal NHPs, albeit informative in that they give a series of basic parameters to assess LV infection and transgene expression in the CNS, do not go beyond the methodology. The most exciting part of the paper comes at the end with the evaluation of beneficial effects of LV-GALC injection in the GCL NHP model. Not only this part of the study is totally novel but also has a strong potential for clinical application. I do realize that the availability of this animal model is very limited, but, as the authors state in their discussion, expanding the analysis of this model after LV-GALC intracerebral injection could give a wealth of information on the enzyme kinetics in circulation and in the CSF, on how efficiently it is taken up by different neural cells, on how it behaves towards galactocerebroside and psychosine buildup in different brain regions and so on. It is a pity that the authors dismiss this part of the paper as simply a proof-of-principle study.

They could have actually split the paper in two, one part describing the methodology and the parameters used to assess the outcome, as proof-of-principle, and the other describing the use of this approach for treatment of a neurodegenerative disease. Right now the paper is very long, descriptive and for the most part methodological.

Points of Concerns

- The fact that the authors found more than 4000 integration sites, mostly within genes implies potential deregulated expression of a large number of functional neural genes. This could have detrimental consequences long-term and should at least be discussed.

- The persistent inflammation at the surgical site and the consistent infiltration of perivascular mononuclear cells should also be assessed long term, especially considering that the authors propose this procedure as safe and applicable for the treatment of pediatric patients.

- Since the authors monitored the treated animals for only three months post surgery, they should be more cautious in excluding a priori the occurrence of vector silencing, or immune response against the transgene or the virus. For the same reason, they should give a more careful assessment of the apparent lack of pathologic effects at the site of focal lesions around the injection site.

- The authors should also clarify what they intend for cytoplasmic distension of neuron-like cells due to "eosinophilic material" which they claim should correlate with high protein expression of both GFP and ARSA?.

Minor points:

- Immunofluorescence staining of oligodendrocytes in Fig 2B is impossible to detect.

- Clarify the WB of GFP in Fig. 3B the molecular weight of the bands differs in slice #3 and slice #9 and SC.

Referee #3 (Comments on Novelty/Model System):

The manuscript, Vasco Meneghini et al;

Pervasive supply of therapeutic lysosomal enzymes in the CNS of normal and Krabbe-affected non-human primates by intracerebral lentiviral gene therapy reports on the in vivo gene therapy by intracerebral (intraparenchymal) delivery of GFP, arylsulfatase A (ARSA) and Galactosylceramidase (GALC) with lentiviral vectors (LVs) in non-human primates (NHPs). The study relies on the hypothesis that intracerebral LV delivery may complement or is alternative to transplantation of autologous hematopoietic stem cell (HSCs) engineered by LVs to deliver relevant enzymes in lysosomal storage disease (LSD) deficiencies (Biffi A, et al (2013) Lentiviral hematopoietic stem cell gene therapy benefits metachromatic leukodystrophy. *Science* 341: 1233158). The study also refers to other in vivo gene therapy protocols for the treatments of severe neurodegenerative LSD disorders in which the approach is based on Adeno-associated viral (AAV) vectors (intracerebral AAV-mediated gene delivery of a functional ARSA enzyme in late infantile MLD; ClinicalTrials.gov Identifier: NCT01801709). The study among other aspects addresses the issue of rapid transgene expression ensured by in vivo gene transfer to tackle the early onset and pathological and clinical fast progressing severe forms of neurodegenerative disorders. Indeed, lentiviral hematopoietic stem cell gene therapy approach demonstrated efficacious besides the slow HSCs ARSA associated expression kinetics. The initial testing of the protocol was performed by injecting unilaterally LV.GFP into two sites anterior external capsule (EC) and thalamus/corona radiata with the use of convection enhanced delivery (CED) device. After this first gene reporter study, the authors decided for another combination of target brain sites and volumes/vector doses: injection of LV containing this time the ARSA gene in two sites anterior EC and thalamus or posterior EC, respectively (Table 1). After assessment of transduction characteristics of LV containing ARSA in NHPs, the authors moved to treat Krabbe disease spontaneous rhesus monkey model for an initial assessment of safety and

efficacy of the proposed protocol. In this case the brain target sites are the internal capsule and the thalamus without the use of convection enhanced delivery (CED) and another volume/vector dose. Therefore the study proceeds with distinct transgenes (GFP to ARSA and finally GALC) and distinct doses routes/doses/volumes. In this respect I would suggest to try to tell a more consistent story in particular the rationale that support and justify the protocol changes from one set of experiments to another.

As it is the manuscript is scattered and not always well organized and consistent. Indeed, the study looks focused on MLD, but with a too explicit drawback for full assessments, which is stated in the abstract: "Given the unavailability of a NHP model of MLD, we delivered the human ARSA transgene in normal juvenile NHPs by using a lentiviral vector developed for ex vivo gene therapy clinical trial". The following sentence: "Then, to validate this strategy in a relevant disease setting, we administered a lentiviral vector encoding for the human GALC transgene in the unique rhesus macaque model of Krabbe disease that recapitulates the human pathology". However, Krabbe disease is not MLD and the protocol ("the strategy") used in the Krabbe rhesus animal model is different from that used in WT NHPs with LV containing ARSA (designed for MLD). Thus, I would suggest keeping a clearer distinction among treatments shown.

"This localized transduction established a far more widespread expression of the enzymes throughout the CNS and in CSF", respect to? This sentence in the abstract dose not state respect to what is "far more widespread expression".

The authors should cross compare in the discussion the levels of increase of ARSA activity measured in the cerebrospinal fluid (CSF) of LV.hARSA-injected NHPs with that published Biffi A, et al (2013) Lentiviral hematopoietic stem cell gene therapy benefits metachromatic leukodystrophy. Science 341: 1233158 and also refer this measure to those published in the following studies.

Intracerebral Gene Therapy Using AAVrh.10-hARSA Recombinant Vector to Treat Patients with Early-Onset Forms of Metachromatic Leukodystrophy: Preclinical Feasibility and Safety Assessments in Nonhuman Primates. Zerah M, et al. Hum Gene Ther Clin Dev. 2015 Jun;26(2):113-24.

Comparative efficacy and safety of multiple routes of direct CNS administration of adeno-associated virus gene transfer vector serotype rh.10 expressing the human arylsulfatase A cDNA to nonhuman primates. Rosenberg JB, et al. Hum Gene Ther Clin Dev. 2014 Sep;25(3):164-77.

Indeed, ARSA activity measured in the cerebrospinal fluid (CSF) appears a convenient parameter to estimate the differences between different protocols and vector platforms.

In Figure 2 a uninjected control should be shown.

Fig S2C some values pre-surgery are missing, without references values it is hard to determine variations.

Fig 3A should be enlarged it is difficult as it is to appreciate the synaptic expression.

Fig 4C in the contralateral hemisphere in group 2 overexpression is not evident

Fig 5 Fig 5 e S5 justify why a difference is present between protein expression and ARSA activity.

In the Krabbe model why sacrifice at 3 months? the key parameter of survival would have been very valuable.

Fig 10 the panels appear with different pattern, are they inverted?

Dear Dr. Buccione,

Enclosed please find the revised version of the manuscript "*Pervasive supply of therapeutic lysosomal enzymes in the CNS of normal and Krabbe-affected non-human primates by intracerebral lentiviral gene therapy*" (EMM-2015-05850).

We thank the reviewers for their overall positive evaluation of our study and for their helpful comments. We edited text and figures trying to address the issues raised by the reviewers, in particular the common points of concern mentioned by reviewers 2 and 3:

1- *The manuscript lacks organization and consistency.* We have now revised and re-focused the manuscript trying to highlight the following concept: the extensive study in normal NHPs (which is now reduced in its methodological part) and the preliminary study in the Krabbe model (which we now describe in a more articulate manner) are complementary and instrumental to the final goal of the work, that is showing the therapeutic potential of the LV-mediated platform to address CNS pathology in LSDs, focusing on MLD and GLD.

Following the reviewers' suggestions we have introduced two main changes:

- we have included a novel paragraph at the beginning of the result section, explaining the rationale of the study, the choice of the experimental models (limitation and advantage) and the motivation of the protocol changes.
- we have re-organized the layout of the results, keeping a clear distinction between treatments, which are now presented as two main sections with subheadings: 1) proof-of-principle of the methodology and parameters used to assess the outcome (LV.hARSA in normal NHPs); 2) therapeutic assessment of the platform in the disease model (LV.hGALC in Krabbe NHP).

2- *The clinically relevant treatment of the Krabbe NHP with LV.hGALC is not considered sufficiently developed.* We are totally aware of the limited set of data. The restricted availability of Krabbe

NHPs allowed treating only one affected animal, which represents however the first report of LV-mediated intracerebral GT described in a NHP model of lysosomal storage disorders. Also, we had to face the very strict rules regulating (basically hampering) the import of NHP tissue samples from the USA to Italy. For these reasons, we do not have additional samples available to perform other analysis, i.e. psychosine or storage quantification. On the available lysates we performed DEAE chromatography, which confirmed rescue of GALC activity in CNS tissues and indicated superimposable biochemical features between the transgenic and the wt enzyme. Of note, all the molecular and biochemical analysis on the Krabbe samples have been performed using the very same standardized protocols and reagents used for LV.ARSA-injected NHPs, thus allowing a direct comparison of results.

Despite the limited sample size and even in the presence of differences in the experimental set up between the two models our results clearly show a remarkable comparable pattern of LV transduction and enzyme biodistribution in juvenile LV.ARSA and LV.GALC-injected NHPs, strongly supporting the consistency of the platform.

Thus, we believe that the comprehensive study performed in normal NHPs and the totally novel pilot study in the disease NHP model complement each other in a full story and provide unique proof-of-concept for efficacy and safety of the LV-mediated intracerebral GT platform.

The abstract and introduction have been revised. Other changes throughout the text and in the figure order and layout have been detailed in the point-by-point reply (below).

Revised text is in red font.

We include a point-by-point reply to the reviewer's comments.

Please note that we are preparing the files containing the whole integration sequence data set that will be submitted to GenBank. The accession code will be provided as soon as the upload is completed.

We thank you for giving us the possibility to revise this manuscript and look forward to your reply.

Point-by-point reply to reviewer's comments

Referee #1

In the present manuscript, Meneghini and collaborators performed a pre-clinical study in non-human primates to assess the biodistribution, transgene expression, immune/inflammatory response, safety parameters and a pilot study on potential therapeutic benefit in one Krabbe-affected animal. Overall this paper provides a strong basis for the potential use of LV for intracerebral administration of ARSA and GALC. The methods are appropriate and well described, the conclusions are balanced and adequately supported by the data. In the discussion, some of the limitations of the present study are clearly mentioned and discussed.

We thank the reviewer for the overall positive evaluation of our work.

1) AAV-ARSA for the treatment of MLD patients is currently ongoing, it is unclear whether the present experimental setting will provide similar/additional therapeutic benefits?

The clinical trial based on AAV.hARSA treatment is ongoing and, to our knowledge, no preliminary data related to short- or medium-term follow up have been released so far. By comparing pre-clinical data obtained in rodents and NHPs using AAV and LV (our strategy) we envisage that LV-based experimental setting might provide equal/increased enzymatic activity in CSF (and consequently in CNS tissues) and direct transduction of oligodendroglial cells without the risk of immune response to the vector and transgene in the absence of immunosuppression regimen, which is normally included in AAV-based treatments.

2) What are the indications that MRI scan is a suitable method for the evaluation of the transduced area? Was this correlated with post-mortem analysis?

We performed MRI scans on treated NHPs immediately post-surgery mainly to assess the consistency of our experimental setting in terms of: a) injection coordinates; b) effective delivery of vector suspension (no bubbles or reflux along the cannula); safety (no hemorrhages or macroscopic tissue damage). Then, we exploited the possibility to estimate from the MRI images the brain volume occupied by the vector suspension. This calculation further supported the consistency of our delivery system. A direct correlation between the injected volume estimated short-term from 3D brain rendering and the transduced volume calculated from vector copy number data obtained 3 months post-surgery would be inappropriate. However, post-mortem molecular and immunohistochemistry analysis, which returned the most informative data regarding the transduced volume, showed indeed the highest vector copy number and the strongest transgene expression (mRNA and protein; GFP and ARSA) in regions close to the injection sites, in close correspondence with MRI images (see **Figure 2, Figure 3, Figure 4, Figure EV1, Appendix Figure S3, Appendix Figure S4**).

3) The authors mentioned that LV are transducing oligodendrocytes but the importance of this specific feature is not discussed in details.

We have added a sentence commenting the importance of transducing oligodendrocytes using LV (**Discussion, pag.14, second paragraph**)

4) Based on the NHP data it is not really possible to determine the added value of CED for LV

delivery. Has a side-by-side comparison of the two delivery systems being performed?

We did not perform a side-by-side comparison (in the same model) of CED versus standard delivery. Indirect evidence that CED enhances the transduction outcome comes from the comparison of data from LV.GFP- and LV.ARSA-injected animals (n=8) with those of LV.GALC-injected NHPs (n=2). Indeed, VCN assessed in regions close to the injection sites is one/two logs higher in CED-injected animals as compared to animals injected with the standard delivery. However, we have to consider that LV.GALC-injected animals received approximately one third of the total TU as compared to the other experimental cohort. More likely, CED favors the distribution of LV particles, enhancing physical diffusion (driven by the positive pressure) along white matter tracts (**Discussion, pag.14, second paragraph**).

CED-mediated and standard delivery systems were used in the study of *Rosenberg et al., Hum Gene Therapy Clinical Dev. 2104* to deliver AAV.hARSA in normal NHPs. However, even in this case a side-by-side comparison is difficult, because the two delivery systems were used to target different regions (deep grey matter using CED and deep grey matter with overlying white matter using non CED-mediated delivery).

5) Figure 3: the information provided by the pictures is extremely limited. High magnification pictures of the transduced areas and ARSA expression should be provided as supplemental Figures.

We have included confocal z-stack pictures showing LV.ARSA-transduced oligodendrocytes (**Figure 2C**) and high magnification pictures showing ARSA expression close to the injection site and in matched tissue blocks of the contralateral hemisphere (**Appendix Figure S3, NEW**).

6) The term "infants" is used (page 16) to describe the study in Krabbe-affected monkey: Juvenile primate would be more appropriate.

We have corrected the terms "infants" with "juvenile" in the text describing Krabbe-affected NHPs (**Results, pag. 10-13; discussion, pag. 17**).

Referee #2

The study is based on the experimentation of intracerebral injection of lentiviral vectors expressing therapeutic enzymes in NHP models, one of which is a naturally occurring model of the lysosomal storage disorder Globoid Cell Leukodystrophy or GLC. These studies are meant to serve as preclinical assessment of the efficacy of this therapeutic approach for the treatment of this devastating disease in affected children.

Remarks:

This paper gives a detailed account of an in vivo gene therapy study targeting primarily the CNS and performed in non-human primates (NHP) models. The study was designed to assess the therapeutic potential of two lentiviral (LV) vectors expressing the lysosomal enzymes arylsulfatase A (ARSA) and galactocerebrosidase (GALC) for the treatment of two severe neurodegenerative, demyelinating, lysosomal storage diseases (LSDs), MLD and GLC, respectively.

In recent years considerable progress has been made towards the development of therapeutics intended to correct or at least ameliorate the CNS disease in LSDs, and several clinical trials have started. However, effective treatment of the aforementioned and other neurodegenerative LSDs continue to be a big challenge for investigators and clinicians alike, and even in cases of ongoing clinical trials long-term patients follow-up are not yet available. In this respect, the current paper could be a valid contribution for both the scientific and clinical community, being the first report on LV-mediated intracerebral gene delivery in an NHP model of a lysosomal disease. Throughout the experimental work has been conducted meticulously and accurately, including the necessary controls. The authors assessed volume and site of injection of the virus at two specific brain locations, chosen to allow for maximal spread of the viral particles; post LV injection, they determined the VCN, the levels of transgene RNA and protein expression across the injected and contralateral brain hemispheres, the spinal cord, the sciatic nerve and peripheral organs. They also verified the number of viral integration sites in brain tissue isolated from the injection site. Overall, the analyses are sound and carefully done.

What in my opinion is difficult to reconcile is the choice of NHP used in this study and the order in which the two main objectives (therapeutic use of LV-ARSA and LV-GALC) of the study are presented.

This taking into consideration that this group of investigators has previously contributed several studies evaluating different therapeutic strategies in models of MLD and GLC, including combination therapy consisting of intracerebral injection of LV-GALC vector and BMT in the Twitcher mouse model of GLC. The same group has also reported on the outcome of a clinical trial conducted in two MLD patients, one pre-symptomatic and the other at the onset of the symptoms. In this comprehensive study they described the beneficial effects of transplanting LV-modified autologous HSCs. Interestingly, they measured high levels of ARSA in the CSF of the treated patients and persistent expression of the therapeutic enzyme in hematopoietic cells with no evidence of clonal expansion of transduced cells. Very impressive results that will become a total break through if the patients will continue to do well in the long run.

Now, as rationale for embarking in the current study the authors argue that ex vivo BM gene therapy might not be sufficiently beneficial for post symptomatic patients given the lag of time potentially needed for CNS repopulation of BM derived cells carrying the therapeutic enzyme. This in spite of the high activity of enzyme measured in the CSF after HSCT, which should be indicative of high levels of circulating enzyme in the brain suited for cross correction of affected neural cells. However, the elaborate pilot studies based on the use of LV-ARSA in normal NHPs, albeit informative in that they give a series of basic parameters to assess LV infection and transgene expression in the CNS,

do not go beyond the methodology. The most exciting part of the paper comes at the end with the evaluation of beneficial effects of LV-GALC injection in the GCL NHP model. Not only this part of the study is totally novel but also has a strong potential for clinical application. I do realize that the availability of this animal model is very limited, but, as the authors state in their discussion, expanding the analysis of this model after LV-GALC intracerebral injection could give a wealth of information on the enzyme kinetics in circulation and in the CSF, on how efficiently it is taken up by different neural cells, on how it behaves towards galactocerebroside and psychosine buildup in different brain regions and so on. It is a pity that the authors dismiss this part of the paper as simply a proof-of-principle study.

They could have actually split the paper in two, one part describing the methodology and the parameters used to assess the outcome, as proof-of-principle, and the other describing the use of this approach for treatment of a neurodegenerative disease. Right now the paper is very long, descriptive and for the most part methodological.

We thank the reviewer for the overall positive evaluation of our present and past work.

We regret that the manuscript in its original form did not convey the message of the study. We have now revised and re-focused the manuscript trying to highlight the following concept: the extensive study based on LV.hARSA injection in normal NHPs and the preliminary study based on LV.hGALC injection in Krabbe NHPs are complementary and instrumental to the final goal of the work, which is showing the therapeutic potential of the LV-mediated platform to address CNS pathology in LSDs.

To address the reviewer's reasonable remarks regarding the rationale of the study, the choice of the experimental models and the order in which results are presented, we have added a paragraph in the result section (**Results, pag. 5, Rationale of the study**). Also, we have re-organized the presentation of the results, keeping a clear distinction between treatments, which are now presented as two main sections with subheadings.

We would like to underline some concepts, some of which are now better specified in the manuscript, as indicated below:

- We have been interested over the past years in defining novel gene/cell therapy strategy for MLD and GLD. Indeed, we gave PoC of safety and efficacy of intracerebral LV.GALC and LV.ARSA delivery in GLD and MLD mice, respectively, with superimposable results in terms of transduction efficiency, vector/transgene distribution and enzymatic reconstitution in CNS tissues (Lattanzi et al., Hum Gene Therapy 2010, 2104) (**Results, page 5; Rationale of the study**).
- Our previous studies of intracerebral GT in MLD and GLD mice (Neri et al., Stem Cells 2011; Lattanzi et al., Hum Gene Therapy 2010, 2104) and in particular the recent study showing the synergic effect of intracerebral GT and BMT in the severe and rapidly progressing GLD mouse model (Ricca et al., Hum Gene Therapy 2015) clearly show that high enzymatic activity is necessary but not sufficient to counteract the disease progression if it is not provided in the early asymptomatic stage and with the appropriate timing to all affected tissues. Indeed, storage, neuroinflammation, CNS and PNS damage are present well before the onset of symptoms in GLD mice as well as in GLD and MLD patients (unfortunately MLD mice do not recapitulate the severity of the human pathology) (**Introduction, pag.3, second paragraph**).
- The short-term follow up (2 years) of the HSC GT clinical trial in 3 infantile/early juvenile MLD patients treated before the onset of symptoms showed that therapeutic benefit and rescue of CNS pathology is correlated to physiological/supraphysiological ARSA activity (up to 2-fold the normal levels) in the CSF (Biffi et al. Science 2013). The medium-term follow up of the same patients and the short-term follow up of additional patients clearly suggest that treating patients before the onset of symptoms is crucial to obtain substantial therapeutic benefit. Indeed, when HSC GT is performed in very early symptomatic patients the rapid CNS disease progression is hardly counteracted, likely because of the time required to donor-derived myeloid progeny to

repopulate the CNS providing therapeutic ARSA activity (Biffi, Naldini, unpublished data). Thus, rapid enzymatic coverage of CNS tissues is essential for these patients.

- Our earlier studies in mice showed that steady levels of transgene expression in CNS tissues and CSF are reached as soon as 5 days after LV-mediated intracerebral GT and are maintained up to one year (the longest time point analyzed), due to the rapid and efficient transduction of neurons and glial cells, that serve as endogenous supply of therapeutic enzyme. Here we give PoC that this is happening in the large brains of NHPs with only two unilateral injections of low LV doses. Due to the obvious limitations in sample size related to NHP studies, we chose 90 days as the only time point for analysis of the LVARSA- and LV.GALC-treated NHPs. However, results of the two LV.GFP-treated NHPs clearly show remarkable steady levels of transgene expression and distribution one month after LV injection, and we envisage that similar levels are reached even at earlier time points. These results provide a solid rationale to propose this approach to stabilize CNS damage and prevent further deterioration in the late onset forms of MLD but also to counteract the rapid progression of CNS pathology in early symptomatic infantile/early juvenile MLD and GLD patients (**Discussion, pag. 14 last paragraph and pag. 15, first paragraph**).
- A NHP model of MLD is not available and LV.ARSA delivery can only be tested in normal NHP. However, we chose to treat juvenile NHPs, in order to assess vector and transgene distribution in an experimental setting that is as close as possible to the potential clinical setting. (**Results, page 5; Rationale of the study**).
- This approach would of absolute importance in GLD, given the severe and even more dramatic disease progression experienced by GLD affected children. Our pilot study in the juvenile Krabbe NHP is extremely important not only because it is the first report of LV-mediated intracerebral GT described in this model, but also because despite some differences in the experimental procedure (which was certainly suboptimal in LV.hGALC-treated NHPs, because we could not use CED and we injected 1/3 of the vector dose as compared to LV.hARSA) we demonstrate quite comparable pattern of LV transduction and enzyme biodistribution in the two models. Thus, we believe that the detailed study performed in normal NHPs goes beyond the methodology and, coupled to the pilot studies in the disease model, represents a real proof-of-concept for efficacy and safety of the LV-mediated intracerebral GT platform (**Results, page 5, Rationale of the study**).

As the reviewer points out, the availability of Krabbe NHPs is extremely limited and we could treat only one affected animal. In addition we had to face the very strict rules regulating the import of NHP tissue samples from the USA to Italy. For these reasons, we do not have additional samples available to perform other analysis, i.e. psychosine quantification. However, we used available lysates to perform DEAE chromatography, which confirmed rescue of GALC activity in CNS tissues and indicated superimposable biochemical features between the transgenic and the wt enzyme (**Figure 7C, NEW**).

We plan to treat other animals using CED and optimized LV doses and injection sites; this will require a joint effort between our group and Prof. Bunnell's group to obtain dedicated funding.

Point of concerns

The fact that the authors found more than 4000 integration sites, mostly within genes implies potential deregulated expression of a large number of functional neural genes. This could have detrimental consequences long-term and should at least be discussed.

The issue of potential gene deregulation due to LV integration is now addressed in discussion (**pag. 16, second paragraph**)

The persistent inflammation at the surgical site and the consistent infiltration of perivascular mononuclear cells should also be assessed long term, especially considering that the authors

propose this procedure as safe and applicable for the treatment of pediatric patients.

Since the authors monitored the treated animals for only three months post surgery, they should be more cautious in excluding a priori the occurrence of vector silencing, or immune response against the transgene or the virus. For the same reason, they should give a more careful assessment of the apparent lack of pathologic effects at the site of focal lesions around the injection site.

We agree that inflammation and infiltration of perivascular mononuclear cells might be a point of concern in the perspective of clinical development of this strategy. We did a careful assessment of pathological lesions in LV.GFP- and LV.hARSA-injected NHPs, which were scored by trained and expert pathologists at MirCen (Fonteney aux Rose, Paris), where the NHPs were treated. Inflammation was always restricted to the injection sites and was scored as mild/moderate in most of the treated animals, with only one animal showing severe infiltration of mononuclear cells. As discussed in the text (**Discussion, pag. 15, last paragraph and pag.16, first paragraph**), pathological scores were not directly correlated to immune response to LV particles or to the human transgene. Importantly, this inflammatory status did not result in neurological manifestations, as assessed by neurobehavioral analysis (**Appendix Figure S2**).

We do not have long-term observation of treated NHPs. Due to the obvious limitations in sample size related to NHP studies, we chose 90 days as the only time point for analysis of the LVARSA- and LV.GALC-treated NHPs, a time in which steady levels of transgene expression and distribution are present. However, our results are similar to those reported in earlier pre-clinical studies in NHPs as well as follow-up studies of gene therapy-treated patients. The issue of immune response is difficult to address in AAV gene therapy studies because treated NHPs as well as patients undergo immunosuppressive regimen. However, these studies report a safe profile of the GT treatment in terms of neuroinflammation. For example, 10 year follow up study of AAV gene therapy in Canavan patients shows no chronic inflammation and no persistent tissue damage (Leone et al., Sci Tr Med 2012). Similarly, one year follow up of study of AAV gene therapy in MPSIIIA children show no sign of inflammation or necrosis (Tardieu et al., Hum Gene Ther. 2014).

In the ProSavin pre-clinical study (LV driving expression of dopaminergic genes) astrogliosis and microgliosis are limited to the vicinity of the injection area 2 and 4 months after vector injection. Inflammation persists up to 84 months post GT in one LV-treated NHP, without evidence of transgene silencing (Jarraya et al. Sci Tr Med 2009). Also, the follow up study of ProSavin gene therapy in PD patients shows no immune response to the transgenes and only transient immune response to LV particles (max 6 months post GT) (Palfi et al., Lancet 2014).

Based on these published data, on the results reported in this study in which we treated juvenile NHPs and on our previous extensive studies performed in neonatal and adult mice (Lattanzi HMG 2010, 2014; Ricca et al. HMG 2015) we envisage that LV silencing and immune response to transgenes are unlikely to occur, even though we cannot exclude them *a priori*. Similarly, neuroinflammation might occur, but will likely be transient and not causing permanent tissue damage (**Discussion, pag. 15, last paragraph and pag.16, first paragraph**).

The authors should also clarify what they intend for cytoplasmic distension of neuron-like cells due to "eosinophilic material" which they claim should correlate with high protein expression of both GFP and ARSA?

The term "eosinophilic material" was used by the pathologist responsible for the histopathological analyses. The exact sentence in the Anatomic Pathology Report was:

"Furthermore, cells with mainly neuronal appearance showed minimal to moderate cytoplasmic distension by homogenous eosinophilic material, central chromatolysis and lateral nuclear displacement in grey matter, in immediate vicinity of inflammatory lesions. These changes were interpreted as intracytoplasmic accumulations of protein. This eosinophilic material may correspond to accumulation with human arylsulfatase A in transduced neurons and have to be confirmed by IHC experiments"

We have included a representative picture showing a neuronal cell presenting the described appearance (**Figure EV3D, NEW**). Indeed, Immunofluorescence and immunohistochemistry confirmed ARSA overexpression in neurons close to the injection site (**Figure 2, Figure 4**)

Minor points

- *Immunofluorescence staining of oligodendrocytes in Fig 2B is impossible to detect.*

We have included two z stack confocal pictures in which expression of ARSA in oligodendrocytes (identified by means of APC and CNPase expression) is evident (**Figure 2C, NEW**).

- *Clarify the WB of GFP in Figure 3B the molecular weight of the bands differs in slice #3 and slice #9 and SC.*

The molecular weight of the bands in the different samples appears slightly different because of the imperfect run of samples in the more external lanes. This “smile effect” is visible also in b-actin bands of the same samples, as shown in the original uncut gels that we include for reviewer’s consideration.

This WB has been moved to **Figure EV1, panel B**

Referee #3

The manuscript, Vasco Meneghini et al; Pervasive supply of therapeutic lysosomal enzymes in the CNS of normal and Krabbe-affected non- human primates by intracerebral lentiviral gene therapy reports on the in vivo gene therapy by intracerebral (intraparenchymal) delivery of GFP, arylsulfatase A (ARSA) and Galactosylceramidase (GALC) with lentiviral vectors (LVs) in non-human primates (NHPs). The study relies on the hypothesis that intracerebral LV delivery may complement or is alternative to transplantation of autologous hematopoietic stem cell (HSCs) engineered by LVs to deliver relevant enzymes in lysosomal storage disease (LSD) deficiencies (Biffi A, et al (2013) Lentiviral hematopoietic stem cell gene therapy benefits metachromatic leukodystrophy. Science 341: 1233158). The study also refers to other in vivo gene therapy protocols for the treatments of severe neurodegenerative LSD disorders in which the approach is based on Adeno-associated viral (AAV) vectors (intracerebral AAV-mediated gene delivery of a functional ARSA enzyme in late infantile MLD; ClinicalTrials.gov Identifier: NCT01801709). The study among other aspects addresses the issue of rapid transgene expression ensured by in vivo gene transfer to tackle the early onset and pathological and clinical fast progressing severe forms of neurodegenerative disorders.

Indeed, lentiviral hematopoietic stem cell gene therapy approach demonstrated efficacious besides the slow HSCs ARSA associated expression kinetics.

The initial testing of the protocol was performed by injecting unilaterally LV.GFP into two sites anterior external capsule (EC) and thalamus/corona radiate with the use of convection enhanced delivery (CED) device. After this first gene reporter study, the authors decided for another combination of target brain sites and volumes/vector doses: injection of LV containing this time the ARSA gene in two sites anterior EC and thalamus or posterior EC, respectively (Table 1). After assessment of transduction characteristics of LV containing ARSA in NHPs, the authors moved to treat Krabbe disease spontaneous rhesus monkey model for an initial assessment of safety and efficacy of the proposed protocol. In this case the brain target sites are the internal capsule and the thalamus without the use of convection enhanced delivery (CED) and another volume/vector dose. Therefore the study proceeds with distinct transgenes (GFP to ARSA and finally GALC) and distinct doses routes/doses/volumes.

In this respect I would suggest to try to tell a more consistent story in particular the rationale that support and justify the protocol changes from one set of experiments to another. As it is the manuscript is scattered and not always well organized and consistent. Indeed, the study looks focused on MLD, but with a too explicit drawback for full assessments, which is stated in the abstract: "Given the unavailability of a NHP model of MLD, we delivered the human ARSA transgene in normal juvenile NHPs by using a lentiviral vector developed for ex vivo gene therapy clinical trial". The following sentence: "Then, to validate this strategy in a relevant disease setting, we administered a lentiviral vector encoding for the human GALC transgene in the unique rhesus macaque model of Krabbe disease that recapitulates the human pathology". However, Krabbe disease is not MLD and the protocol ("the strategy") used in the Krabbe rhesus animal model is different from that used in WT NHPs with LV containing ARSA (designed for MLD). Thus, I would suggest keeping a clearer distinction among treatments shown.

We thank the reviewer for the overall positive evaluation of our work.

We regret that the manuscript in its original appeared scattered and did not clearly convey the message of the study. We have now revised and re-focused the manuscript trying to highlight the following concept: the extensive study based on LV.hARSA injection in normal NHPs and the preliminary study based on LV.hGALC injection in Krabbe NHPs are complementary and instrumental to the final goal of the work, which is showing the therapeutic potential of the LV-mediated platform to address CNS pathology in LSDs.

We specifically addressed the reviewer's reasonable remarks regarding the rationale of the study as well as the rationale that supports and justify the protocol in the two experimental models by adding a novel paragraph in the result section (**Results, pag 5, Rationale of the study**). Also, we have re-organized the presentation of the results, keeping a clear distinction between treatments, which are now presented as two main sections with subheadings.

We would like to highlight the following points, some of which are now better specified in the manuscript, as indicated below:

- The clinical trial based on AAV.hARSA treatment is ongoing and, to our knowledge, no preliminary data related to short- or medium-term follow up have been released so far. By comparing pre-clinical data obtained in rodents and NHPs using AAV and LV (our strategy) we envisage that LV-based experimental setting might provide equal/increased enzymatic activity in CSF (and consequently in CNS tissues) and direct transduction of oligodendroglial cells without the risk of immune response to the vector and transgene in the absence of immunosuppression regimen, which is normally included in AAV-based treatments.
- The short-term follow up (2 years) of the HSC GT clinical trial in 3 infantile/early juvenile MLD patients treated before the onset of symptoms showed that therapeutic benefit and rescue of CNS pathology is correlated to physiological/supraphysiological ARSA activity (up to 2-fold the normal levels) in the CSF (Biffi et al. 2013). In these patients HSC GT approach is efficacious besides the slow HSCs ARSA associated expression kinetics.
The medium-term follow up of the same patients and the short-term follow up of additional patients clearly suggest that treating patients before the onset of symptoms is crucial to obtain substantial therapeutic benefit. Indeed, when HSC GT is performed in very early symptomatic patients the rapid CNS disease progression is hardly counteracted, likely because of the time required to donor-derived myeloid progeny to repopulate the CNS providing therapeutic ARSA activity (Biffi, Naldini, unpublished data). Thus, an approach providing rapid enzymatic coverage of CNS tissues would be essential for these patients.
- Our earlier studies in mice (Lattanzi et al., 2010, 2014; Ricca et al, 2015) showed that steady levels of transgene expression in CNS tissues and CSF are reached as soon as 5 days after LV-mediated intracerebral GT and are maintained up to one year (the longest time point analyzed), due to the rapid and efficient transduction of neurons and glial cells, that become an endogenous supply of therapeutic enzyme.
Here we give proof-of-concept that this is happening in the large brains of NHPs with only two unilateral injections of low LV doses.
- Due to the obvious limitations in sample size related to NHP studies, we chose 90 days as the only time point for analysis of the LVARSA- and LV.GALC-treated NHPs. However, results of the two LV.GFP-treated NHPs clearly show remarkable steady levels of transgene expression and distribution one month after LV injection, and we envisage that similar levels are reached even at earlier time points. These results provide a solid rationale to propose this approach to stabilize CNS damage and prevent further deterioration in the late onset forms of MLD but also to counteract the rapid progression of CNS pathology in early symptomatic infantile/early juvenile MLD and GLD patients **Discussion, pag. 14 last paragraph and pag. 15, first paragraph**).
- A NHP model of MLD is not available and LV.ARSA delivery can only be tested in normal NHPs, which, despite obvious limitation in sample size as compared to rodents, allow some flexibility and optimization of the protocol. Importantly, we chose to treat juvenile NHPs (normal and Krabbe-affected), in order to assess vector and transgene distribution in an experimental setting that is as close as possible to the potential clinical setting (**Results, pag.5 Rationale of the study**).

Specific points:

"This localized transduction established a far more widespread expression of the enzymes throughout the CNS and in CSF", respect to? This sentence in the abstract dose not state respect to

what is "far more widespread expression".

We changed the sentence in "Efficient gene transfer in neurons, astrocytes and oligodendrocytes close to the injection sites resulted in robust production and extensive spreading of transgenic enzymes in the whole CNS and in CSF"

The authors should cross compare in the discussion the levels of increase of ARSA activity measured in the cerebrospinal fluid (CSF) of LV.hARSA-injected NHPs with that published Biffi A, et al (2013) Lentiviral hematopoietic stem cell gene therapy benefits metachromatic leukodystrophy. Science 341: 1233158 and also refer this measure to those published in the following studies:

- Intracerebral Gene Therapy Using AAVrh.10-hARSA Recombinant Vector to Treat Patients with Early-Onset Forms of Metachromatic Leukodystrophy: Preclinical Feasibility and Safety Assessments in Nonhuman Primates. Zerah M, et al. Hum Gene Ther Clin Dev. 2015 Jun;26(2):113-24.

- Comparative efficacy and safety of multiple routes of direct CNS administration of adeno-associated virus gene transfer vector serotype rh.10 expressing the human arylsulfatase A cDNA to nonhuman primates. Rosenberg JB, et al. Hum Gene Ther Clin Dev. 2014 Sep;25(3):164-77.

Indeed, ARSA activity measured in the cerebrospinal fluid (CSF) appears a convenient parameter to estimate the differences between different protocols and vector platforms.

Cross-comparison of ARSA activity in the CSF of LV.hARSA-treated NHPs in our study and HSC gene therapy treated MLD patients indicate comparable increase of enzymatic activity resulting from treatment: in normal LV.ARSA-treated NHPs we reached up to 2-fold the normal levels (100% increase) at 3 months post gene therapy, while in MLD children ARSA activity was rescued to physiological or up to 2-fold the physiological levels (in one patient) at 12 and 24 months after treatment (**Discussion, pag. 15**). Of note, the vector used and the enzymatic assay applied to determine ARSA activity were the same in the two studies.

CSF	Meneghini et al.	Biffi et al.2013
Normal donors (untreated)	2.14 ± 0.47 nmol/mg*h (mean±SEM; n=3) range 1.66 - 3.09	Average ≈ 1.2 nmol/mg*h (n=2)
Before treatment	Physiological	Undetectable
After treatment	4.88 ± 0.89 (mean±SEM; n=5) range 3.99 - 5.78 nmol/mg*h	Average ≈ 1 (n=3) Range ≈ 0.9 – 2.4
Mean increase (fold to normal donors)	≈ 2	≈ 1.5

The levels of ARSA activity in the CSF of AAV.ARSA-treated NHPs were not reported in the studies by Rosenberg et al., 2014 and Zerah et al., 2015.

Taking into account differences in the treatment protocols, we observe overall comparable levels of increased ARSA activity in CNS tissues of treated NHPs between those studies and our study (see graphs below). Please note that these levels were achieved with only two injections in our study, while 6 injection sites (with 6 or 12 deposits) were used to deliver AAV.hARSA in the studies of Rosenberg et al., 2014 and Zerah et al., 2015.

Meneghini et al. *

*data reported in FigureS7A. In order to facilitate comparison with other studies we have pooled data from the injected and contralateral hemispheres.

Rosenberg et al., 2014

Zerah et al., 2015

	NHP1	NHP2
Mean increase in ARSA activity in the injected hemisphere	14%	31%
Percentage of cubes with ARSA overactivity (over endogenous level)		
> 10%	53	57
> 30%	28	35
> 50%	13	21
> 100%	2	9

In Figure 2 an uninjected control should be shown.

We included a representative picture of an untreated control (**Figure 2D, NEW**)

Fig S2C some values pre-surgery are missing, without references values it is hard to determine variations.

The pre-surgery sera were available only for three animals. For this reason a statistical test to compare pre- and post values for each animal was not feasible. The best we could do was to perform a 2way ANOVA with Bonferroni's correction considering treatments (pre- and post-surgery) and analyte as source of variation. The test revealed no significant effect of the treatment on the result ($P=0.21$) and no significant interaction between variables ($P=0.53$).

Fig 3A should be enlarged it is difficult as it is to appreciate the synaptic expression.

We included enlarged pictures showing GFP-expressing cells (soma and neuronal processes) in **Figure EV1**.

Fig 4C in the contralateral hemisphere in group 2 overexpression is not evident

ARSA overexpression in the contralateral hemisphere of group1 and group2 is not appreciable when we quantify immunopositive signal after IHC, because this tool is not sensitive enough to reveal small differences on a physiological background of ARSA activity (**Results, pag.7**).

Fig 5 Fig 5 e S5 justify why a difference is present between protein expression and ARSA activity.

Both Figure 5 and Figure S5 (**now Figure EV2**) refer to ARSA activity. In Figure 5 we show supraphysiological ARSA activity in the injected and contralateral hemisphere, in all the brain slices analyzed. In **Figure EV2** we show that more tissue blocks display supraphysiological ARSA activity

in animals of study group 2, and confirm ARSA overexpression by chromatographic analysis and by measuring ARSA activity towards the natural substrate sulfatide.

The difference between the pattern of protein expression (**Figure 4**) and activity (**Figure 5** and **Figure EV2**) is explained by the fact that even a relatively small pool of LV-transduced ARSA-overexpressing cells produce and secrete relevant amount of functional ARSA enzyme that is secreted and circulate in the CSF (**Figure 4G**), thus being available in regions distant from the injection site.

In the Krabbe model why sacrifice at 3 months? the key parameter of survival would have been very valuable.

We agree that monitoring survival is paramount to demonstrate efficacy. However, since we had only one treated animal, we preferred to evaluate vector and transgene biodistribution at a time point that was comparable with that chosen for the LV.hARSA-treated animals, in order to better evaluate the consistency of the platform.

Fig 10 the panels appear with different pattern, are they inverted?

We presume the reviewer is alluding to **Figure S10 (now Figure EV5)**

Figure S10A panels are correctly displayed

Figure S10B panels are correctly displayed, despite a lower intensity of the signal is evident in the right as compared to the left panel (compare for example signal from Twi UT lanes, in which the same sample was loaded). This is likely due to technical reasons (i.e. exposure time, stripping of the membrane). We include the uncut gels for reviewer's consideration.

Uncut gels in Fig. EV5

Thank you for the submission of your revised manuscript to EMBO Molecular Medicine. We have now received the enclosed reports from the referees that were asked to re-assess it.

As you will see, while Reviewers 1 and 3 are now globally supportive, Reviewer 2 remains unconvinced by your rebuttal and modifications, especially with respect to the perceived lack of complementarity between the two experimental models used and consequent inconclusiveness regarding clinical and translational relevance. After extended internal discussion, and considering that Reviewer 3 had similar concerns and is now satisfied, I can inform you that we will be able to accept your manuscript pending the following final amendments:

1) Please discuss further the other points of concern expressed by Reviewer 2. It would be helpful if you provide a manuscript with all the red lettering removed, plus an additional copy highlighting only the changes introduced as a consequence of this decision letter

2) The "The Paper Explained" section must be incorporated in the main manuscript

3) The manuscript must include a statement in the Materials and Methods identifying the institutional and/or licensing committee approving the experiments, including any relevant details (like how many animals were used, of which gender, at what age, which strains, if genetically modified, on which background, housing details, etc). We encourage authors to follow the ARRIVE guidelines for reporting studies involving animals. Please see the EQUATOR website for details: <http://www.equator-network.org/reporting-guidelines/improving-bioscience-research-reporting-the-arrive-guidelines-for-reporting-animal-research/>. Please make sure that all the above details are reported. In this case, most information is reported but please make sure the age and gender of animals is also reported in the manuscript

4) As per our Author Guidelines, the description of all reported data that includes statistical testing must state the name of the statistical test used to generate error bars and P values, the number (n) of independent experiments underlying each data point (not replicate measures of one sample), and the actual P value for each test (not merely 'significant' or ' $P < 0.05$ ').

5) We are now encouraging the publication of source data, particularly for electrophoretic gels and blots, with the aim of making primary data more accessible and transparent to the reader. Would you be willing to provide a PDF file per figure that contains the original, uncropped and unprocessed scans of all or at least the key gels used in the manuscript? The PDF files should be labeled with the appropriate figure/panel number, and should have molecular weight markers; further annotation may be useful but is not essential. The PDF files will be published online with the article as supplementary "Source Data" files. If you have any questions regarding this just contact me.

6) Please note that we now mandate that all corresponding authors list an ORCID digital identifier. You may do so through our web platform upon submission and the procedure takes <90 seconds to complete. We encourage all authors to supply an ORCID identifier, which will be linked to their name for unambiguous name identification.

7) You are welcome to suggest a striking image or visual abstract to illustrate your article. If you do please provide a jpeg file 550 px-wide x 400-px high.

8) We note that the file sizes for the figures are very high. This will cause in the file transfer to production in case of acceptance. Please endeavor to significantly reduce the file sizes, obviously without compromising the current quality.

Please submit your revised manuscript within two weeks. I look forward to seeing a revised form of your manuscript as soon as possible.

***** Reviewer's comments *****

Referee #1 (Remarks):

The revised version has been substantially modified and restructured to take into account the reviewers' comments.

Referee #2 (Comments on Novelty/Model System):

The experimental work was conducted accurately and thoroughly, including adequate controls. However, the two experimental models used in this study are in my view not complementary to one another, as emphasized in my comments to the authors. For this reason, the potential medical impact is difficult to predict based on the data presented in this paper. I do not believe this revised version of paper is suitable for publication in EMBO Molecular Medicine.

Referee #2 (Remarks):

The authors have attempted to address my major concerns by better explaining the rationale of including two different model systems in their study.

I remain of the idea that the paper as presented, although well done, addresses two, not necessarily complementary, therapeutic questions.

The first is the use of LV-ARSA in wild type NHPs to determine the spreading of the virus and expression of the therapeutic enzyme in the brain. This part of the study could have medical relevance if the authors assume that LV-ARSA injected in wild type NHP and the parameters used to determine its therapeutic efficacy would behave identically in affected children, which of course remains to be seen. However, this large portion of the study is primarily methodological and should be published as a method.

The second, most interesting part of the study regards the correction of the demyelinating disease in the NHP model of Krabbe disease. The use of this model as simply a proof-of-principle model to generalize the use of this therapeutic approach in two clinically distinct LSDs, i.e. MLD and Krabbe disease, is in my view an overstatement.

The study with LV-ARSA cannot be considered complementary to those with LV-GALC. Although these two leukodystrophies may share clinical similarities, they are caused by deficiency of quite different enzymes, ARSA, a sulfatase, and GALC, a glycosphingolipid cleaving enzyme. Therefore, they might present with therapeutic hurdles that are dependent on the biochemical and kinetic properties of the individual enzymes. For this reason I strongly suggest to report these interesting studies separately.

Minor points still of concern:

- 1) The issue of the multiple Insertion sites of the LV is still not well addressed. The authors should be concerned not only with potential LV insertion in oncogenes or genes associated with cell transformation, but also with insertion that would interfere with normal neuronal function.
- 2) In the Western blot now present in Fig. EV1 the difference in molecular weight of the GFP bands cannot be due to imperfect running of the samples, as the authors claim. The inclusion of the original uncut gel makes even more evident that the three bands on the right of the gel have a higher molecular weight.

Referee #3 (Remarks):

The manuscript revised is worthy of publication

Point-by-point reply to Reviewer 2 comments

We thank the reviewer for recognizing the quality of our work.

We respect the reasons of the reviewer, even if we partially disagree. Indeed, we still believe that the two models evaluated in our study provide complementary information and are worth presenting in the same manuscript.

We are aware that despite sharing common pathological traits and having clinical similarities MLD and GLD presents with peculiar features that may impact on the development of effective treatments. We also know from our previous studies that GALC and ARSA have indeed different biochemical properties, possibly different mechanisms of uptake into the cells, and also different mechanisms (likely cell-specific) regulating their expression. Indeed, while stable supraphysiological ARSA levels are well tolerated in HSCs with maintenance of the complex functions of the hematopoietic system (Biffi et al. 2013; Capotondo et al. 2007), toxicity resulting from GALC-overexpression in HSCs requires a complex gene transfer vector design to down-regulate transgene expression in donor HSCs while allowing its (over)expression in the therapeutically-relevant HSC myeloid progeny (Gentner et al. 2011; Ungari et al. 2015). In addition, difficulties to reach supraphysiological GALC levels in these tissues are suggestive of a peculiar molecular and biochemical characteristics of GALC enzyme, with stringent regulation of its expression (Ricca et al. 2015; Visigalli and Biffi 2011).

Despite these differences in the biology of the two systems, the most important requirement for effective treatment of the CNS pathology in both diseases is the achievement of stable, therapeutically relevant levels of enzyme expression/activity in the brain and spinal cord (which may vary but likely have to be close to physiological). Our study provide strong evidence that the proposed LV-mediated GT platform achieves this purpose, as shown by the consistent pattern of vector and enzyme biodistribution observed in LV.hGALC- and LV.hARSA-injected NHPs even in the presence of different transgenes and different in treatment protocols. In particular, the extent of ARSA overexpression in the CSF of normal NHPs (comparable to the physiological ARSA levels detected in the liquor of MLD patients 12-24 months post HSC gene therapy (Biffi et al, 2013)) and of GALC rescue in CNS tissues and CSF the Krabbe model lead to reasonable expectation of therapeutic benefit in patients, which of course remains to be evaluated in rigorous clinical trials.

We have introduced a sentence in Discussion to better address these issues (pag. 16).

Minor points

1. We have better addressed the issue of LV insertion sites, mentioning that LV insertion may potentially interfere with normal cell functions (Discussion, pag. 18)
2. Western blot in Fig. EV1. We are confident that the three bands at the right of the gel identify GFP protein (expected MW 27-28 kDa, but slight shifts are possible, according to the information in the data sheet). Indeed, the antibody is highly specific and there are no confounding bands at higher or lower MW, since GFP is not present in NHP tissues (unless in treated animals). Moreover, imperfect run is evident also in the b-actin bands. In order to provide additional evidence we include below a different

blot in which we ran tissue samples of the same LV.GFP treated NHP (P2). In this blot samples run correctly (but we had a problem with b-actin in lane 2) and, as in the previous one, no other confounding bands are present, as shown in the uncutted gels (below).

We provide this information for the reviewer and editor consideration. We might include a revised version of Fig.1EV including the alternative panel B if required.

Alternative panel B, Fig.1EV

Legend

Representative western blot showing GFP expression in selected tissue blocks of the injected and contralateral hemispheres from brain slices 3, 5, and of SC (blocks #1 and #4) of LV.GFP-injected P2 NHP. β-actin was used as housekeeping gene. Positive controls (lane 1 and 2): LV.GFP-transduced COS7 cells (VCN=1; 1:5 and 1:50 dilution). Negative control: brain tissue from untreated U14 NHP (UT NHP).

Uncut gels for alternative panel B, Fig.1EV

GFP

b-actin

The two lanes at the right side of the gel correspond to brain tissues of untreated, wt mice that were included in this run.

Corresponding Author Name: Angela Gritti
 Journal Submitted to: EMBO Molecular Medicine
 Manuscript Number: EMM-2015-05850-V2